# A chloroplast retrograde signal, 3'-phosphoadenosine 5'-phosphate, acts as a secondary messenger in abscisic acid signaling in stomatal closure and germination

Wannarat Pornsiriwong[1,2†], Gonzalo M Estavillo[1,3†], Kai Xun Chan[1†], Estee E Tee[1], Diep Ganguly[1], Peter A Crisp[1], Su Yin Phua[1], Chenchen Zhao[4], Jiaen Qiu[5,6], Jiyoung Park[7], Miing Tiem Yong[4], Nazia Nisar[1], Arun Kumar Yadav[1], Benjamin Schwessinger[8], John Rathjen[8], Christopher I Cazzonelli[1,9], Philippa B Wilson[1], Matthew Gilliham[5], Zhong-Hua Chen[4,10], Barry J Pogson[1*]

[1]ARC Centre of Excellence in Plant Energy Biology, Research School of Biology, The Australian National University, Acton, Australia; [2]Department of Biochemistry, Faculty of Science, Kasetsart University, Bangkok, Thailand; [3]CSIRO Agriculture and Food, Acton, Australia; [4]School of Science and Health, Western Sydney University, Richmond, Australia; [5]ARC Centre of Excellence in Plant Energy Biology, Department of Plant Science, School of Agriculture, Food and Wine, University of Adelaide, Glen Osmond, Australia; [6]Waite Research Institute, University of Adelaide, Glen Osmond, Australia; [7]Division of Biological Sciences, Cell and Developmental Biology Section, University of California, San Diego, San Diego, United States; [8]Research School of Biology, The Australian National University, Acton, Australia; [9]Hawkesbury Institute for the Environment, Western Sydney University, Richmond, Australia; [10]College of Agriculture and Biotechnology, Zhejiang University, Hangzhou, China

*For correspondence: barry. pogson@anu.edu.au

†These authors contributed equally to this work

Competing interests: The authors declare that no competing interests exist.

**Abstract** Organelle-nuclear retrograde signaling regulates gene expression, but its roles in specialized cells and integration with hormonal signaling remain enigmatic. Here we show that the SAL1-PAP (3'-phosphoadenosine 5'- phosphate) retrograde pathway interacts with abscisic acid (ABA) signaling to regulate stomatal closure and seed germination in *Arabidopsis*. Genetically or exogenously manipulating PAP bypasses the canonical signaling components ABA Insensitive 1 (ABI1) and Open Stomata 1 (OST1); priming an alternative pathway that restores ABA-responsive gene expression, ROS bursts, ion channel function, stomatal closure and drought tolerance in *ost1-2*. PAP also inhibits wild type and *abi1*-1 seed germination by enhancing ABA sensitivity. PAP-XRN signaling interacts with ABA, ROS and $Ca^{2+}$; up-regulating multiple ABA signaling components, including lowly-expressed Calcium Dependent Protein Kinases (CDPKs) capable of activating the anion channel SLAC1. Thus, PAP exhibits many secondary messenger attributes and exemplifies how retrograde signals can have broader roles in hormone signaling, allowing chloroplasts to fine-tune physiological responses.

## Introduction

Organelles such as chloroplasts and mitochondria can regulate nuclear gene transcription *via* several signaling pathways in a process called retrograde signaling (*Chi et al., 2013*; *Chan et al., 2016a*). While this has been conventionally viewed as a bilateral communication to optimize organelle function and/or repair, there is emerging evidence that retrograde signaling contributes to multiple cellular processes and complex whole-plant traits including programmed cell death, drought tolerance, biotic stress tolerance, light-regulated seedling development and flowering (*Chi et al., 2013*; *Chan et al., 2016a*; *Feng et al., 2016*; *Kleine and Leister, 2016*). Interestingly, specific role(s) have never been examined for any chloroplast retrograde signal identified to date in relation to drought tolerance and abscisic acid (ABA) -mediated signaling in specialized cells such as guard cells surrounding stomata. The hormone ABA mediates signaling pathways that regulate stomatal closure and seed germination. The timing of seed germination needs to be coordinated with favorable environmental conditions to ensure seedling viability, while stomata are the gateways for gas exchange and water loss in leaves and thus closure mediated by guard cells is one of the most important and immediate avoidance responses to drought stress in plants (*Murata et al., 2015*).

Intriguingly, although regulation of stomatal closure by ABA directly impacts on photosynthesis and chloroplast function (*Yamburenko et al., 2015*), how and to what extent signals emanating from oxidatively-stressed chloroplasts may be integrated with ABA signaling in guard cells have remained largely enigmatic. The metabolite 3'-phosphoadenosine 5'-phosphate (PAP) acts as a retrograde signal during oxidative stress. PAP accumulates during high light exposure and drought *via* redox inactivation of its catabolic phosphatase SAL1, and moves from chloroplasts to the nucleus *via* a transporter (*Estavillo et al., 2011*; *Gigolashvili et al., 2012*; *Chan et al., 2016b*). PAP is perceived by and inhibits exoribonuclease (XRN)-mediated RNA metabolism as evidenced in *xrn* double and triple mutants phenocopying *sal1* mutants; resulting in drought tolerance and activation of 25% of the high light stress transcriptome. Mutant alleles lacking SAL1 catabolic activity, such as *altered ascorbate peroxidase2 expression 8* (*alx8*, hereafter termed *sal1*-8) and *fiery1*-6 (*fry1*-6, hereafter termed *sal1*-6), constitutively accumulate PAP and are consequently drought tolerant (*Rossel et al., 2006*; *Wilson et al., 2009*; *Estavillo et al., 2011*).

PAP was initially proposed to act largely via ABA-independent pathways as the drought tolerance in *sal1* correlated with accumulation of osmoprotectants, and there were conflicting reports on the impacts of *sal1* mutations on stomatal conductance: an earlier study suggested that SAL1 was not involved in stomatal regulation, whereas we found markedly decreased stomatal conductance in *sal1* with elevated PAP (*Xiong et al., 2001*; *Rossel et al., 2006*; *Wilson et al., 2009*; *Estavillo et al., 2011*). Additionally, a subset of ABA-responsive genes are misregulated in *sal1* mutants (*Wilson et al., 2009*), raising the question as to whether PAP can participate in ABA-mediated processes such as stomatal closure and seed germination.

Binding of ABA to its receptors (RCAR/PYR1/PYL) (*Ma et al., 2009*; *Park et al., 2009*) leads to inactivation of the group A Protein Phosphatase 2C (PP2C) proteins such as ABI1 and activation of SNF1-Related Kinases 2.2, 2.3 and 2.6/OST1 (SnRK2.2, SnRK 2.3, SnRK2.6/OST1) (*Koornneef et al., 1984*; *Leung et al., 1994*; *Meyer et al., 1994*; *Mustilli et al., 2002*). The central role of PP2Cs and SnRKs in ABA signaling are demonstrated by the reduced sensitivity to ABA-mediated germination inhibition and stomatal closure in *abi1*-1 which is insensitive to ABA-PYR/PYL-mediated inhibition, and stomatal closure in *ost1*-2 (*Koornneef et al., 1984*; *Leung et al., 1994*; *Meyer et al., 1994*; *Mustilli et al., 2002*; *Umezawa et al., 2009*). Indeed, both seeds and guard cells of the triple mutant of ABA-activated SnRKs are almost completely ABA-insensitive (*Fujii and Zhu, 2009*; *Nakashima et al., 2009*). In guard cells active OST1 phosphorylates Slow Anion Channel-Associated 1 (SLAC1) allowing anion release, as well as facilitating potassium efflux by stimulating potassium efflux channels and inhibiting the inward potassium channels (KAT1 and KAT2), respectively (*Geiger et al., 2009*; *Sato et al., 2009*; *Brandt et al., 2012*). These anion and cation fluxes are necessary to close stomata (*Schroeder and Hagiwara, 1989*; *Vahisalu et al., 2008*; *Geiger et al., 2009*). OST1 activation also triggers gene expression changes, production of reactive oxygen species (ROS) including hydrogen peroxide ($H_2O_2$) *via* NADPH oxidases, and interacts with intracellular $Ca^{2+}$ signaling which involves cytosolic fluctuations in $Ca^{2+}$ levels termed $Ca^{2+}$ transients (*Murata et al., 2015*).

The ABA-induced intracellular $Ca^{2+}$ transients activate Calcium Dependent Protein Kinases (CDPKs) (*Mori et al., 2006*). There are at least 34 CDPKs in *Arabidopsis thaliana*, of which at least eight function in ABA signaling and ROS homeostasis in guard cells (*Boudsocq and Sheen, 2013*; *Zou et al., 2015*; *Simeunovic et al., 2016*). The subgroup II CDPKs are particularly important in guard cell signaling because three members (CDPK3, 21 and 23) can regulate SLAC1 and KAT channel activities (*Cheng et al., 2002*; *Geiger et al., 2010*; *Brandt et al., 2012*, *2015*). With respect to SLAC1; CDPK3, 21 and 23 preferentially phosphorylate a residue different from that targeted by OST1 (*Geiger et al., 2010*; *Brandt et al., 2012*, *2015*). Thus, SLAC1 channel activity is controlled by the joint action of OST1 and three CDPKs in counterbalance with competitive dephosphorylation by PP2Cs (*Brandt et al., 2015*).

The understanding of guard cell regulation is far from complete, in part due to the complex interaction between ABA and secondary messengers such as $Ca^{2+}$ and ROS. Notably, $Ca^{2+}$-activation of CDPKs and ROS production by NADPH oxidases are interdependent processes with multiple layers of feedback regulation. Furthermore, the extent to which SnRK2-independent ABA/$Ca^{2+}$signaling contributes to stomatal closure in whole plants under drought stress has not been thoroughly investigated, as various reports to date have largely utilized epidermal peels of unstressed plants or heterologous systems (*Geiger et al., 2010*; *Brandt et al., 2012*, *2015*). In this regard, how ABA, $Ca^{2+}$ and ROS in guard cells may interact with chloroplast signaling during drought stress also remains to be elucidated.

Herein we present the unexpected finding that a chloroplast-mediated retrograde signaling pathway can bypass the canonical ABA pathway described above, which closes stomata and restores responsiveness in *abi1*-1 and *ost1*-2 mutants to ABA thereby conferring drought tolerance to these hitherto drought-sensitive mutants. The novel roles of the nucleotide phosphatase SAL1 and its associated phosphoadenosine signal, PAP, in guard cell regulation are investigated in the context of known and uncharacterized ABA signaling components and secondary messengers. Finally, the extent to which the interaction between ABA and PAP signaling occurs in other tissues and processes, such as seed germination, was also investigated.

## Results

### PAP restores guard cell responsiveness in ABA-insensitive plants

We investigated whether SAL1-PAP can interact with ABA signaling by crossing the *sal1* mutant, *sal1*-8, with the ABA insensitive mutants *abi1*-1 and *ost1*-2 (*Koornneef et al., 1984*; *Leung et al., 1994*; *Mustilli et al., 2002*). Elevated levels of PAP, a lack of SAL1 protein and many of the visible phenotypes of *sal1*-8 were still present in the double mutants (*Figure 1A*, *Figure 1—figure supplement 1A*). When challenged by drought stress, the single mutants, *abi1*-1 and *ost1*-2, displayed the expected widespread wilting and death before wild type. Conversely and unexpectedly, the ABA signaling double mutants containing the *sal1*-8 lesion were green, turgid and photosynthetically viable after ten days of drought (*Figure 1A*).

We then studied whether the drought tolerance in *abi1*-1 *sal1*-8 and *ost1*-2 *sal1*-8 was due to restoration of guard cell ABA responsiveness. The SAL1 protein is present in epidermal peels and localized to chloroplasts of guard cells (*Figure 1—figure supplement 1B,C*). Significantly, while the stomatal conductance ($g_s$) in *ost1*-2 *sal1*-8 and *abi1*-1 *sal1*-8 remained high under control conditions, the double mutant guard cells exhibited restored ABA sensitivity, with their $g_s$ and stomatal aperture decreasing in response to ABA (*Figure 1B,C*). Consistent with this finding, higher PAP content in well-watered *sal1*-8 plants conferred constitutively decreased $g_s$ and elevated leaf temperature which are indicative of enhanced stomatal closure [*Figure 1B*, *Figure 1—figure supplement 2B* and *Rossel et al. (2006)*]. Both epidermal peels and leaves of intact *sal1*-8 plants also exhibited enhanced sensitivity to ABA and closed stomata to a greater extent than wild type (*Figure 1—figure supplement 2*).

### Complementation in ABA insensitive mutants by PAP is not due to changes in stomatal index, genetic background or ABA content

We investigated whether the complementation in *abi1*-1 *sal1*-8 and *ost1*-2 *sal1*-8 is driven by restoration of ABA signaling by PAP as opposed to contribution from other non-specific or pleiotropic

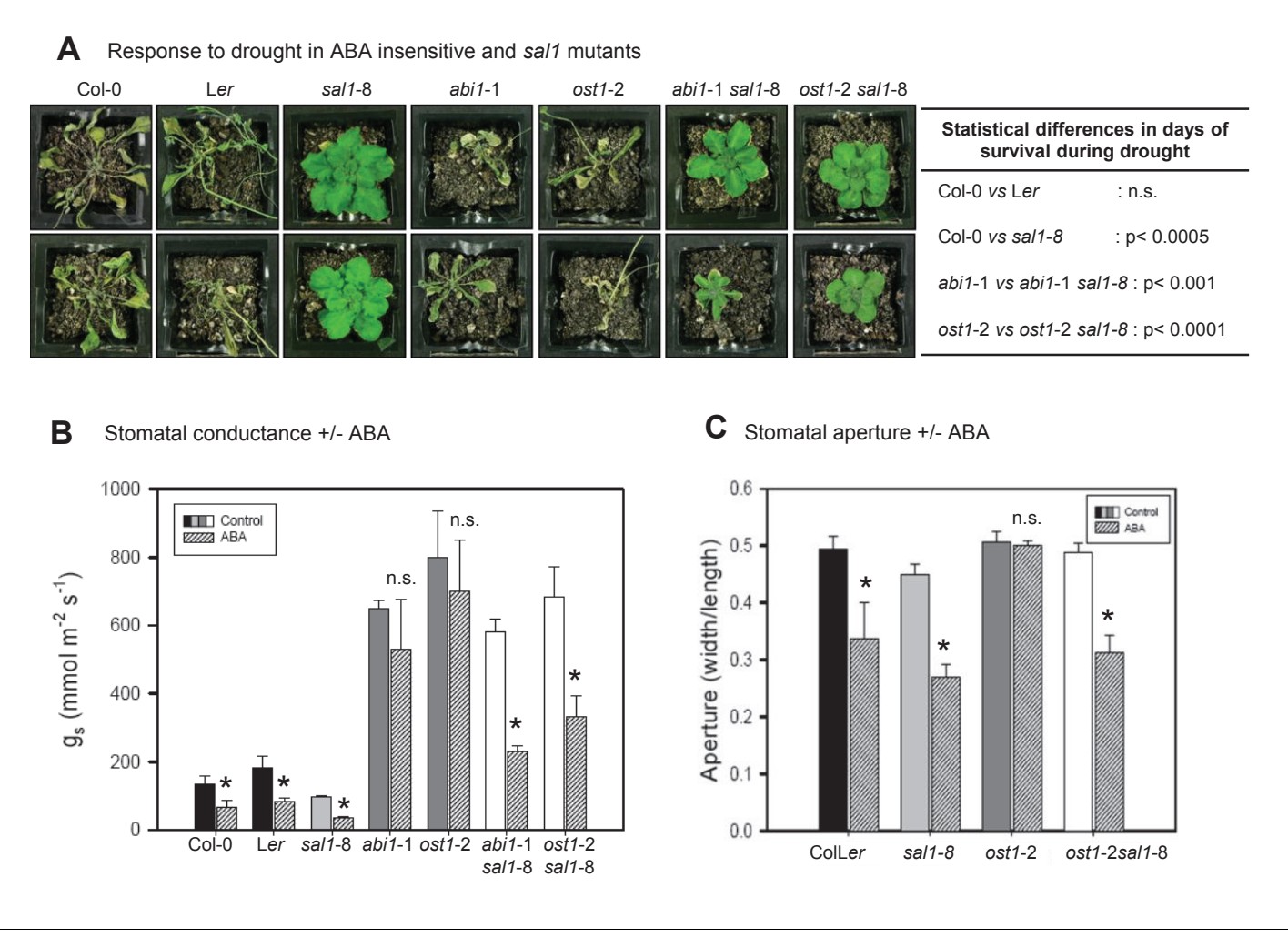

**Figure 1.** PAP restores drought tolerance and ABA-responsive stomatal closure in ABA signaling mutants. (**A**) Representative photos of two plants per genotype subjected to 10 days of drought. Statistically significant differences in survival during drought between genotypes are indicated (n = 4 per genotype per experiment, three independent experiments). (**B**) Effect of 20 µM ABA on stomatal conductance ($g_s$) after 2 hr feeding through the roots of hydroponically-grown plants. The data is the average of two independent experiments (n = 3 plants per genotype per experiment) ± SEM. (**C**) The effect of 50 µM ABA treatment for 2 hr on stomatal aperture of leaf peels from five to six-week old plants.

The following figure supplements are available for figure 1:

**Figure supplement 1.** PAP content in single and double mutants; and SAL1 localization in epidermal peels and guard cell chloroplasts.

**Figure supplement 2.** Enhanced ABA sensitivity in guard cells of *sal1*-8.

**Figure supplement 3.** The biochemistry, physiology and stomatal features of single and double mutants, and testing of PAP action *via* pathogen signaling.

effects. We previously showed that drought tolerance in *sal1*-8 is not due to slower water loss from soil (*Wilson et al., 2009*) and the tolerance is conferred by PAP in shoots, not roots (*Hirsch et al., 2011*). Therefore we tested whether PAP in leaves of the double mutants decreased constitutive $g_s$ or stomatal density, which can enhance drought tolerance (*Doheny-Adams et al., 2012*; *Hepworth et al., 2015*). The $g_s$ in well-watered double mutants were still as high as those in the parental *abi1*-1 and *ost1*-2 (*Figure 1B*). CryoSEM measurements of stomatal morphology and density revealed that the significantly higher total stomatal opening area per leaf area in the ABA

signaling single mutants (*abi1*-1, *ost1*-2) remained high in the drought tolerant double mutants and were not decreased to wild type levels by *sal1*-8, which itself has wild type-like stomatal opening area per leaf area (*Figure 1—figure supplement 3A*). Next, since *ost1*-2 and *abi1*-1 are in L*er* and *sal1*-8 in Col-0, we analyzed Col-0, L*er*, ColL*er* F1 hybrids and segregating F2 and F3 plants of the crosses; no ecotype effects that could account for the drought tolerance independent of the *sal1*-8 mutation were observed (*Figure 1—figure supplement 3A,B*). We also generated double mutants containing both lesions in the Col-0 background using T-DNA mutants of *ost1* (*salk_008068*) and *sal1*-6. The *ost1* (Col-0 background) mutant was similar to *ost1*-2 (L*er* background), being ABA-insensitive and failing to close stomata after four days of drought stress (*Figure 1—figure supplement 3C,D*). Significantly, the *ost1 sal1*-6 (Col-0 background) mutant had restored stomatal closure under drought stress (*Figure 1—figure supplement 3D*), ruling out ecotype effects as the major driver for the drought tolerance in *ost1*-2 *sal1*-8.

We then tested whether the complementation could be explained by differences in ABA content. We previously reported that ABA levels are increased in *sal1*-8 (*Rossel et al., 2006*) and herein observed that ABA content was indeed slightly higher in *sal1*-8 and *ost1*-2 *sal1*-8, but this elevation was not significantly different (ANOVA, p=0.09) (*Figure 1—figure supplement 3E*). However, the marginally higher ABA content did not decrease $g_s$ values in the well-watered double mutants, which were still as high as those in the parental *abi1*-1 and *ost1*-2 (*Figure 1B*). Furthermore, when ABA-deficient *aba2*-3 [a leaky allele (*Léon-Kloosterziel et al., 1996*; *Laby et al., 2000*; *Barrero et al., 2005*)] was crossed to *sal1*-8, the double mutant showed no significant change in relative water content after eight days of water stress (WW 0.81 ± 0.11 vs WS 0.84 ± 0.06), as opposed to a significant decline for *aba2*-3 (WW 0.78 ± 0.02 vs WS 0.61 ± 0.08, p<0.05). Correspondingly, *aba2*-3 *sal1*-8 survived significantly longer than *aba2*-3 (16 days vs 11 days, p<0.005) as assayed by chlorophyll fluorescence (*Woo et al., 2008*). Therefore the restoration of ABA responsiveness and drought tolerance in *abi1*-1 *sal1*-8 and *ost1*-2 *sal1*-8 by PAP did not appear to primarily proceed *via* enhanced ABA synthesis, nor is it likely to, given the extensively reported insensitivity of *abi1*-1 and *ost1*-2 to ABA.

## Biochemical manipulation of PAP signaling induces stomatal closure, and PAP-mediated stomatal closure can be enhanced by ABA, but does not act via flg-22 mediated pathogen signaling

We hypothesized that if PAP is a genuine signal regulating stomatal closure and it interacts with ABA signaling, then application of exogenous PAP to leaves should elicit similar responses as other known guard cell regulators such as ABA, ROS and Ca$^{2+}$. Therefore we established and validated protocols for direct PAP application to leaves either *via* petioles or application to epidermal leaf peels; and evaluated effectiveness, uptake, transport and degradation of the fed PAP. In our system both barley and *Arabidopsis* leaf peels responded to the positive control, ABA, to a degree expected for each species compared to the mock measuring buffer containing Ca$^{2+}$[which is known to promote certain levels of stomatal closure (*Blatt et al., 1990*)]. We then tested 10, 50 and 100 μM exogenous PAP. The PAP-induced closure, shown for 100 μM (*Figure 2A,B*) was significantly greater than the mock. Both 10 and 50 μM PAP were capable of causing a similar degree of closure to 100 μM PAP (10 μM PAP: 59 ± 5% closure, 50 μM PAP: 52 ± 7%, 100 μM PAP: 46 ± 8%; p=0.4 by ANOVA), albeit at a slower rate as expected for a physiological dose-dependent response. Significantly, both the rate and extent of closure of Arabidopsis and barley leaf peels with 100 μM PAP was comparable to the respective ABA responses (*Figure 2A,B*). We then tested whether exogenous PAP could induce stomatal closure in *ost1*-2, and observed significant PAP-induced closure (72 ± 2% closure in +PAP *vs* 90 ± 1% in control, p<0.001).

Next we investigated the uptake, transport and degradation of exogenous PAP in guard cells by biochemically manipulating its transport and degradation in leaf peels and in petiole-fed leaves. Exogenous ATP is a known co-substrate for the PAP transporter (*Gigolashvili et al., 2012*), therefore we used 10-fold higher ATP as this is predicted to outcompete PAP for import into chloroplasts, thus preventing the degradation of exogenous PAP. In leaf peels, applying ATP and PAP simultaneously increased the rate and magnitude of stomatal closure (*Figure 2C*). We then measured and observed significantly elevated PAP content in petiole-fed leaves co-treated with PAP+ATP compared to +PAP or +ATP alone (*Figure 2—figure supplement 1*). The elevated PAP in PAP+ATP leaves was associated with an increase in leaf temperature (*Figure 2D*), a typical consequence of

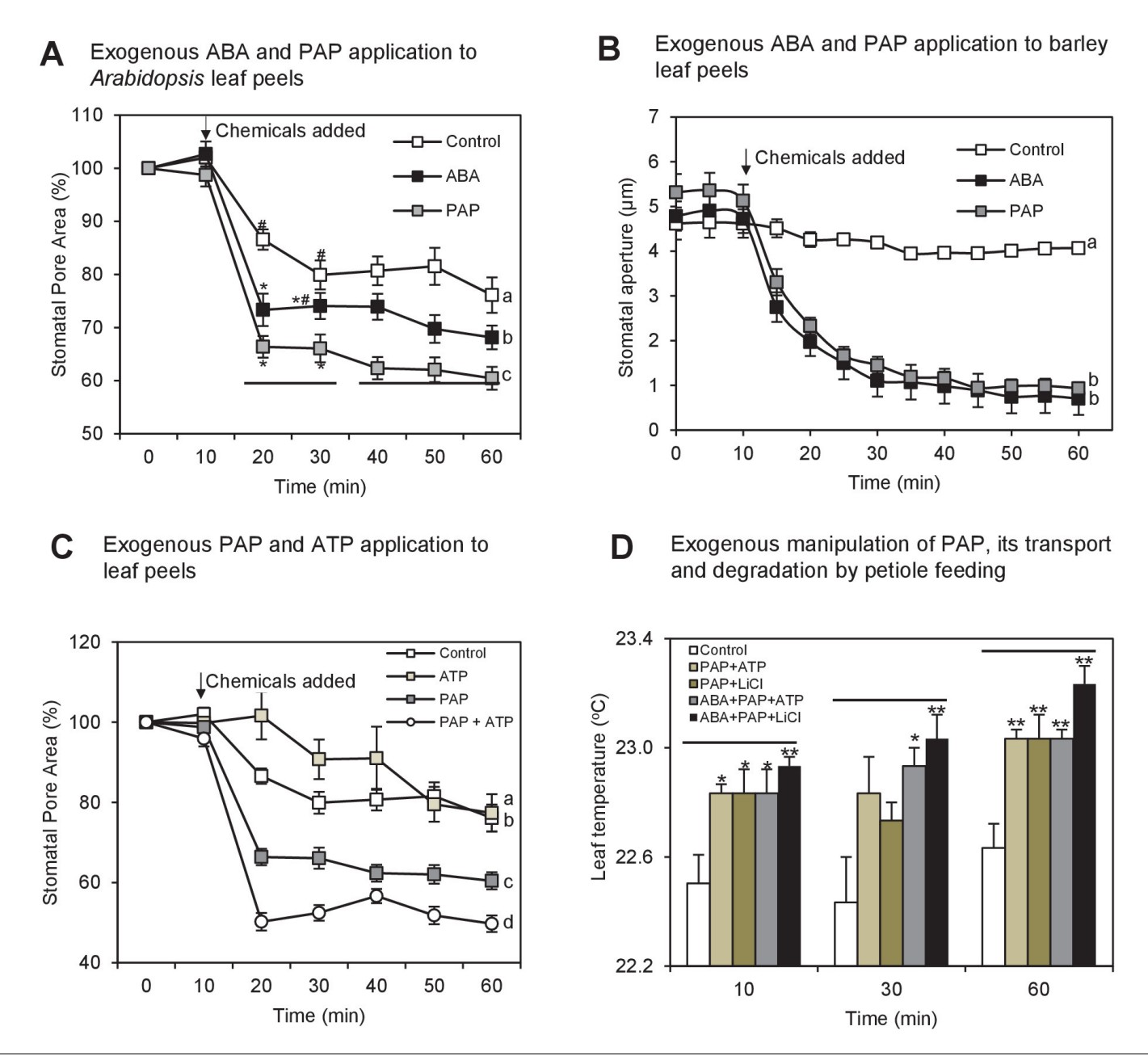

**Figure 2.** Exogenous PAP interacts with ABA signaling and acts in stomatal closure in both *Arabidopsis* and barley. (A) Stomatal aperture, calculated using measurements of pore width and length, in leaf peels of wild type (Col*Ler*) plants treated with either 100 µM PAP or 100 µM ABA over a period of 1 hr. Values are means, expressed as a percentage compared to t = 0 min, of at least 20 stomata ± SEM. Rates of closure were compared by modelling the closure between 10–25 min (log-transformed data), significant difference groups (p<0.05) are denoted by #, *. Final level of closure was also considered by ANOVA across the final 30 min; significant difference (p<0.05) denoted a, b, c. (B) Stomatal aperture in leaf epidermal peels of three-week old barley plants in measuring buffer (Control) for 10 min before adding 100 µM ABA or 100 µM PAP for another 50 min. Values are means ± SEM (n = 17–20 stomata of four plants). Significant difference (p<0.05) is denoted a, b. (C) Stomatal aperture as in (A) but treated with either 100 µM PAP or 1 mM ATP alone or in combination, in measuring buffer. Values are means of at least eight stomata ± SEM. The control treatment for (A), (B) and (C) was 1 hr of measuring buffer. (D) Thermography of 35-day old wild type leaves petiole fed with 250 µL of different combinations of 20 µM ABA, 100 mM LiCl, 1 mM PAP and 10 mM ATP in infiltration buffer or buffer alone (Control). Mean and SEM of leaf temperature from three leaves from three plants per genotype are shown. Leaves in solution were returned to growth chamber and temperature measured at indicated timepoints. Significant differences to control are shown (*p<0.05; **p<0.01). Also see *Figure 2—figure supplement 1*.

The following figure supplement is available for figure 2:

*Figure 2 continued on next page*

*Figure 2 continued*

**Figure supplement 1.** Exogenous PAP feeding to plant leaves *via* epidermal leaf peels or petiole feeding.

stomatal closure. We then tested for PAP degradation in petiole-fed leaves by using LiCl, a SAL1 inhibitor (*Quintero et al., 1996*), and observed elevated PAP with a similarly increased leaf temperature (*Figure 2D*, *Figure 2—figure supplement 1*). Leaf temperature by PAP+LiCl could be further enhanced by co-treatment with ABA (*Figure 2D*), suggesting synergistic action of PAP accumulation and ABA.

A theoretical explanation for our observation that exogenous PAP closes stomata is that it is perceived apoplastically and then stimulates a chloroplast calcium response that can close stomata, as is the case for the flg-22 Pathogen Associated Molecular Pattern (PAMP) pathogen response pathway mediated by the chloroplastic Calcium-Sensing Receptor (CAS) (*Nomura et al., 2012*; *Guzel Deger et al., 2015*). However, we found that PAP-mediated stomatal closure does not require functional CAS protein (*Figure 1—figure supplement 3F*), and, in agreement with the published data neither does ABA (*Han et al., 2003*; *Nomura et al., 2008*). We then tested whether PAP elicits an extracellular PAMP response by feeding PAP, ABA or the flg22 PAMP receptor elicitor to leaf disks. Luminol-based assays of leaf disks revealed that the magnitude and kinetics of ROS burst in the ABA- or PAP-treated leaf disks were 40-fold lower than that induced by flg22 (data not shown).

## PAP restores multiple ABA signaling outputs in ABA-insensitive plants

Given the connections between PAP and ABA responses we systematically tested for the complementation of key outputs of ABA signaling, namely ion channel fluxes, ROS bursts and gene expression, in wild type, *ost1*-2 *sal1*-8 plants and in *ost1*-2 plants treated with exogenous PAP. First, $K^+$ and $Cl^-$ fluxes which are indicators of ion transport into or out of guard cells that enable stomatal closure were measured. Experiments with ion-selective microelectrodes revealed that ion fluxes in *ost1*-2 were not responsive to ABA treatment, but ion fluxes of *ost1*-2 *sal1*-8 were restored to wild type and *sal1*-8 levels, with $K^+$ and $Cl^-$ efflux both stimulated upon ABA treatment (*Figure 3A*). This indicates complementation of the *ost1*-2 phenotype in *ost1*-2 *sal1*-8, most likely through genetically accumulated PAP (*Figure 1—figure supplement 1A*). Exogenous PAP also stimulated the ion flux responses from guard cells similar to that of ABA and $Ca^{2+}$ in all genotypes tested (*Figure 3A,B*; *Figure 3—figure supplement 1A*). However, exogenous PAP does not affect the activity of the key ion channels SLAC1, KAT1 and KAT2 when supplemented into oocytes (*Figure 3—figure supplement 1B*); suggesting that PAP influences the ion fluxes indirectly through restoration of ABA signaling.

Second, the ROS burst in guard cells measured with the 2',7'-dichlorodihydrofluorescein diacetate ($H_2$DCFDA) dye was observed in response to ABA treatment in wild type and was attenuated as expected in *ost1*-2 (*Figure 3C*). The $H_2$DCFDA dye primarily detects $H_2O_2$ and to a lesser extent hydroxyl radicals (*Wojtala et al., 2014*). No significant photooxidation of $H_2$DCFDA was observed for the control measuring buffer treatment within the timeframe of the experiment (mean fluorescence of $1.2 \pm 0.4 \times 10^5$ at 0 min vs $1.5 \pm 0.5 \times 10^5$ at 10 min, p=0.37). Significantly, the ABA-mediated ROS burst was restored in ABA-treated *ost1*-2 *sal1*-8, to a similar degree to that of wild type. Furthermore, the timing and extent of the ABA-induced ROS burst was phenocopied by exogenous PAP in wild type, *ost1*-2 and *ost1*-2 *sal1*-8 (*Figure 3C*).

Third, we performed transcriptome analyses of well-watered whole leaves of wild type, *ost1*-2, *sal1*-8, and *ost1*-2 *sal1*-8 plus or minus ABA (*Figure 3D–E*, *Figure 3—figure supplement 2* , *Supplementary file 1*). As expected, *ost1*-2 gene expression profiles were largely unresponsive to ABA compared to wild type leaves. Significantly, the transcriptional response to ABA was substantially restored in *ost1*-2 *sal1*-8 +ABA although the magnitude was attenuated (*Figure 3D*, *Supplementary file 1*). Specifically, for a subset of 1723 genes that responded differently to ABA in wild type compared to *ost1*-2, the expression of 1705 (99%) genes was largely restored in the *ost1*-2 *sal1*-8 plus ABA, such that they were no longer significantly different to wild type plus ABA (*Supplementary file 1*, FDR adjusted p<0.05). Because the transcriptome was performed on whole leaves, we verified whether complementation was also observed in *ost1*-2 *sal1*-8 for 1173 ABA-

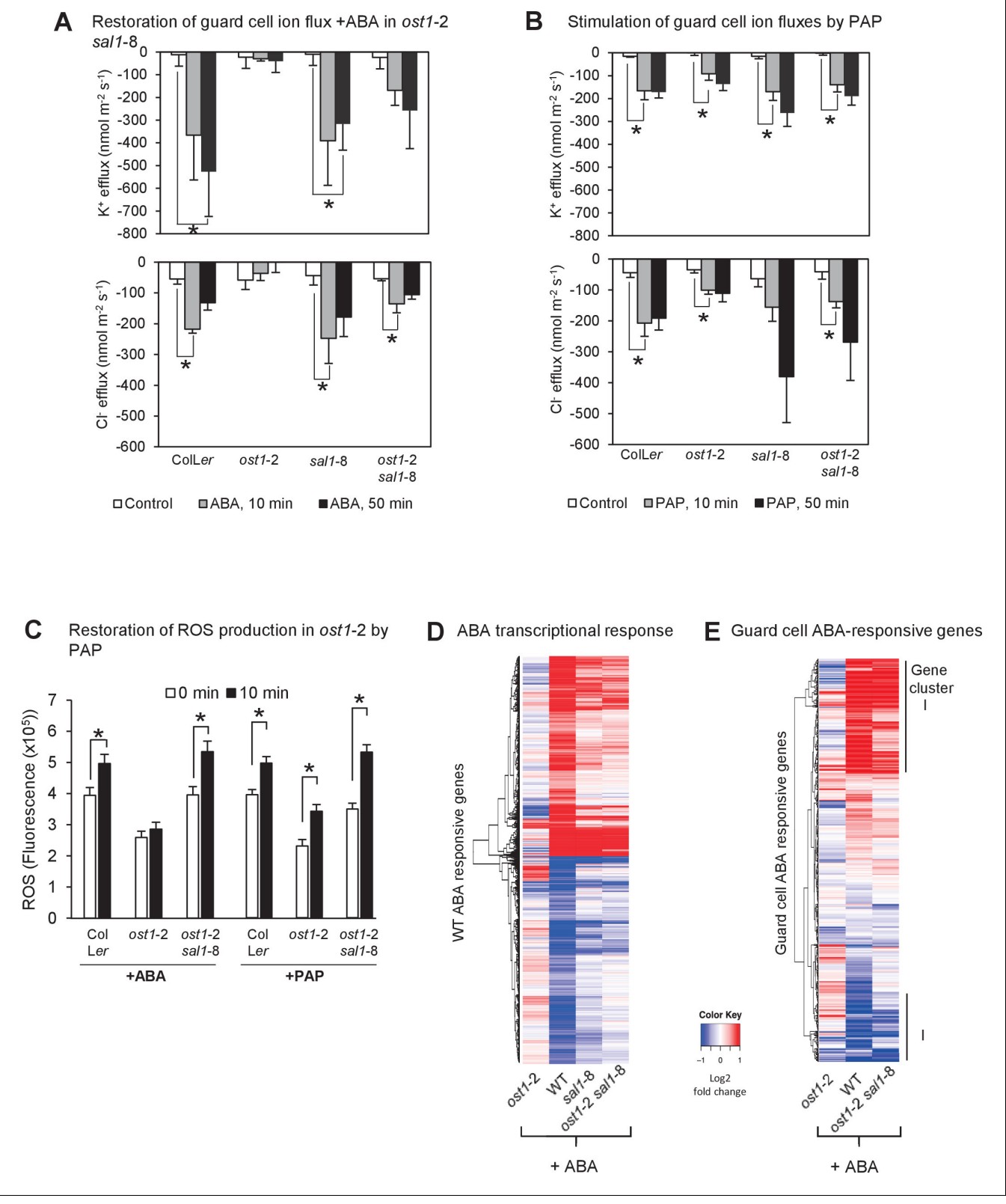

**Figure 3.** PAP restores guard cell ion fluxes, ROS burst and global transcriptional response to ABA when accumulated genetically in *ost1-2 sal1-8* and when applied exogenously to *ost1-2* and wild type. Effects of (**A**) 500 μM ABA or (**B**) 500 μM PAP on combined net flux of each of the ion transporters for K⁺ and Cl⁻ from guard cells in leaf epidermal peels of four-week old Arabidopsis plants. Average net ion fluxes ± SEM (n = 5–7 plants) are shown for control, 10 min and 50 min after ABA or PAP treatment. Asterisk shows statistically significant difference to 0 min (p<0.05, ANOVA). (**C**) Mean

*Figure 3 continued on next page*

*Figure 3 continued*

corrected total cell fluorescence of ROS in the presence of 2′,7′-dichlorodihydrofluorescein diacetate (H₂DCFDA), a ROS probe that detects primarily H₂O₂ and to a lesser extent hydroxyl radicals (*Wojtala et al., 2014*), in guard cells before and after 10 min of 100 μM ABA or 100 μM PAP. Means ± SEM of 45–54 (+ABA) or 73–92 (+PAP) stomata per genotype is shown with significant differences denoted (t-test, p<0.05). (D–E) Hierarchical clustering comparing ABA transcriptional response in wild type (WT) and mutants for (D) transcripts responsive to ABA in WT in this study; and (E) transcripts known to respond to ABA in guard cells (*Wang et al., 2011*). The blue (down)-red (up) scale is log₂ fold change for each genotype +/− ABA, respectively. The scale has been condensed such that the red and blue colours at the end of the scale encompass all fold-changes greater or equal to 2, or less than or equal to 0.5, respectively. Clusters showing co-expression in WT and *ost1-2 sal1-8* are marked (I). Also see *Supplementary file 1*.

The following figure supplements are available for figure 3:

**Figure supplement 1.** Effect of endogenously accumulated and exogenous PAP on ion fluxes in guard cells and transporter activity in oocytes respectively.

**Figure supplement 2.** Changes in gene expression of ABA receptors, PP2Cs and SnRKs in response to ABA.

responsive genes expressed in guard cells (*Wang et al., 2011*). Indeed, a large proportion of these guard cell-expressed genes that were induced by ABA in wild type were not induced in *ost1*-2, but were significantly induced in *ost1*-2 *sal1*-8 (gene cluster I, *Figure 3E*).

We re-analyzed the transcriptome using the 132 genes known to be involved in ABA signaling (*Hauser et al., 2011*). Hierarchical clustering revealed some subsets differentially down-regulated in all *sal1*-8 backgrounds and treatments compared to wild type +ABA, and a few genes more highly induced by ABA in *sal1*-8 and *ost1*-2 *sal1*-8, but not in *ost1*-2 (*Figure 3D*). Significantly, we identified several genes involved in diverse, but interlinked, aspects of ABA signaling that were differentially expressed in *ost1*-2 *sal1*-8 under constitutive and/or +ABA conditions. This included two transcription factors, 16 Ca²⁺ signaling proteins [CDPKs, CRKs, CBLs and CIPKs], and genes regulating ROS homeostasis as well as vesicle trafficking to plasma membranes (*Supplementary file 2*). Furthermore, many of these genes up-regulated in *ost1*-2 *sal1*-8 have been shown to have the ability to regulate the activities of inward rectifying channels (KATs) and slow anion channels (SLAC1/SLAHs) for stomatal closure. For instance, CDPK19-mediated signaling is required for regulation of potassium inward currents by ABA and Ca²⁺ (*Zou et al., 2015*), CIPK6 directly activates the key K⁺ channel AKT1 (*Lan et al., 2011*), while various up-regulated CDPKs are closely related to the group II CDPKs (CDPK3, 21 and 23) known to directly regulate SLAC1 and SLAH3 activity (*Geiger et al., 2010*; *Brandt et al., 2012*). Two up-regulated vesicle transport syntaxin genes, SYP41 and SYP124, also have protein-protein interactions and close sequence relationships with syntaxins known to regulate K⁺ channels and stomatal closure (*Sanderfoot et al., 2001*; *Eisenach et al., 2012*).

Collectively, these results indicate that PAP-mediated signaling restores ABA sensitivity at multiple levels for stomatal closure in plants lacking wild type OST1 or ABI1. If so, how is PAP perceived in the cell during ABA signaling and how does it act?

## Investigation of mechanisms by which PAP-mediated retrograde signaling might intersect with ABA signaling to regulate stomatal closure

To investigate the possibility that PAP perception and signaling in ABA responses act *via* its established retrograde signaling pathway we crossed *ost1*-2 to the drought tolerant *xrn2*-1 *xrn3*-3 double mutant, which genetically phenocopies PAP inhibition of nuclear XRNs and *sal1* (*Dichtl et al., 1997*; *Estavillo et al., 2011*). Similar to observations obtained for *ost1*-2 *sal1*-8, ABA responsiveness and stomatal closure were restored in both intact plants and epidermal leaf peels of *ost1*-2 *xrn2*-1 *xrn3*-3 (*Figure 4A,B*). ABA-responsive guard cell ion fluxes were restored to wild type levels in *ost1*-2 *xrn2*-1 *xrn3*-3, whereas *ost1*-2 remained insensitive as expected (*Figure 4C*). We also observed complementation of ABA-responsive ROS burst in this triple mutant to the same extent as that seen in *ost1*-2 *sal1*-8 (*Figure 4D*). Therefore, our data indicate that PAP participates in ABA signaling through its established SAL1-PAP-XRN retrograde communication pathway.

To investigate the mechanism(s) by which PAP-mediated retrograde signaling restores ABA sensitivity in *ost1*-2, we first tested whether functionality of the PYR1/PYL/RCAR-mediated ABA signaling

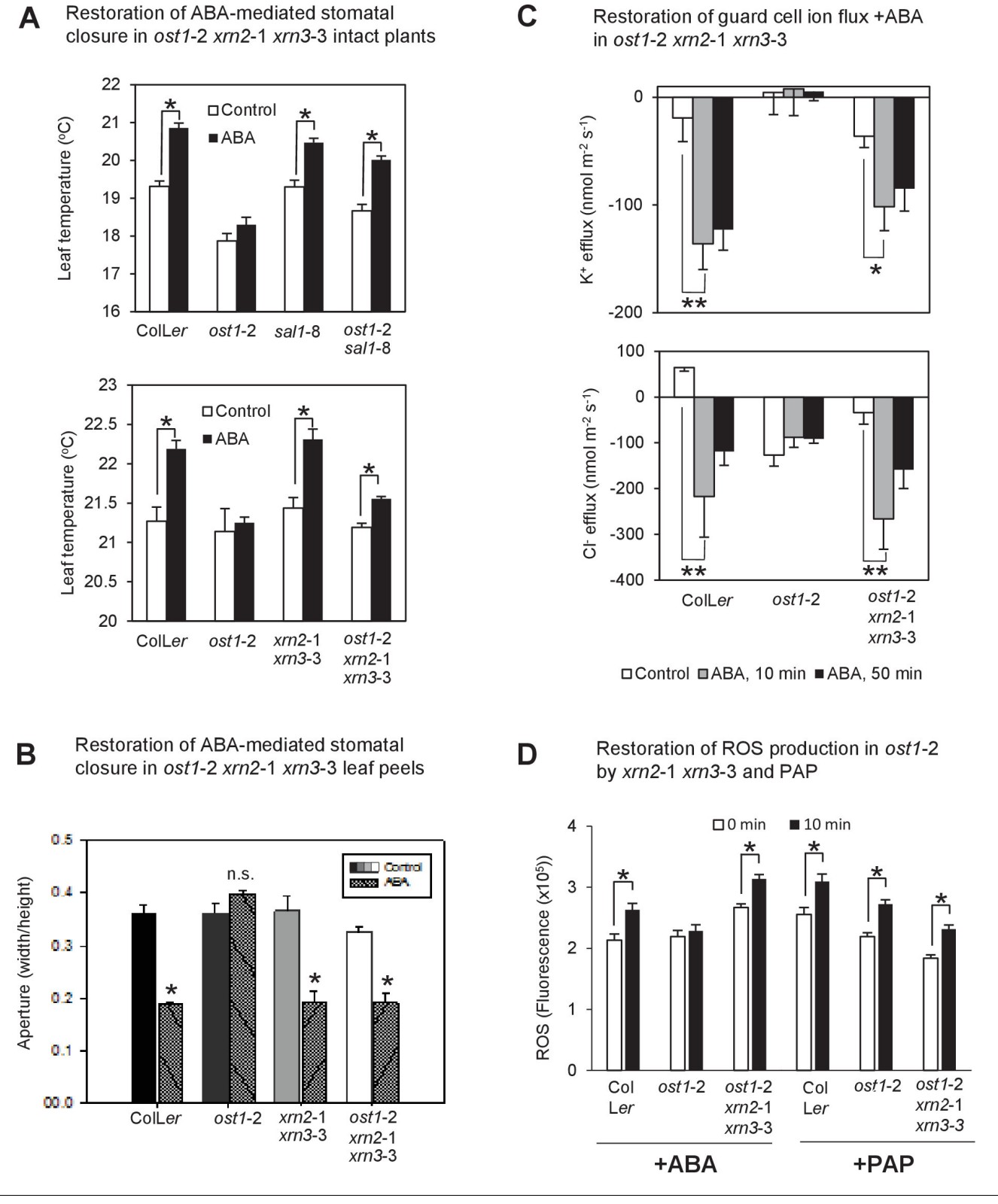

**Figure 4.** Restoration of ABA-responsive stomatal closure, guard cell ion fluxes and ROS production in *ost1-2 xrn2-*1 *xrn3-*3. (**A**) Leaf temperature, a proxy of stomatal closure, in leaves of intact plants of wild type, *ost1-2*, *sal1-8*, *ost1-2 sal1-8*, *xrn2-*1 *xrn3-*3 and *ost1-2 xrn2-*1 *xrn3-*3 after treatment with 20 µM ABA for 2 hr. Significant differences relative to control are indicated by asterisk (*p<0.05). Values shown are means ± SEM of 3–9 biological replicates per treatment. The ABA responses of *ost1-2 sal1-8* and associated controls were performed in an earlier, independent experiment compared

*Figure 4 continued on next page*

*Figure 4 continued*

to *ost1*-2 xrn2-1 xrn3-3. (B) Stomatal aperture in leaf peels of wild type, *ost1*-2, xrn2-1 *xrn3*-3 and *ost1*-2 xrn2-1 *xrn3*-3 after treatment with 50 µM ABA for 2 hr (40–60 stomata per genotype ± SD). Significant difference (*) at p<0.05 is shown for +ABA. The same trend was observed in two independent experiments. (C) Effects of 500 µM ABA on combined net flux of each of the ion transporters for K$^+$ and Cl$^-$ from guard cells in leaf epidermal peels of four-week old Arabidopsis plants. Average net ion fluxes ± SEM (n = 4–12 plants) are shown for control, 10 min and 50 min after ABA treatment. Significant differences are indicated by asterisks (*p<0.05, **p<0.01; ANOVA) (D) Mean corrected total cell fluorescence of ROS in the presence of 2',7'-dichlorodihydrofluorescein diacetate (H$_2$DCFDA), a ROS probe that detects primarily H$_2$O$_2$ and to a lesser extent hydroxyl radicals (*Wojtala et al., 2014*), in guard cells before and after 10 min of 100 µM ABA or 100 µM PAP. Means ± SEM of 110–190 (+ABA) or 130–200 (+PAP) stomata from four biological replicates per genotype are shown with significant differences denoted (t-test, p<0.05).

cascade is restored in *ost1*-2 *sal1*-8 using ABA analogues. The ABA analogue, 3'-ethylsulfanyl-ABA (AS2), is a limited-spectrum ABA agonist. AS2 activates the dimeric ABA receptors PYR1, PYL1, PYL2 and PYL3 to comparable levels as ABA to initiate the signaling cascade that leads to stomatal closure; also partially activates the monomeric receptors PYL4, PYL5 and PYL11; but cannot activate PYL6, PYL9 and PYL10 (*Takeuchi et al., 2014*). AS2 induced stomatal closure to a similar extent as ABA in both wild type and *ost1*-2 *sal1*-8, while *ost1*-2 did not respond to either AS2 or ABA as expected (*Figure 5A*). Hence, ABA signaling and downstream molecular responses mediated through dimeric, and possibly some monomeric, ABA receptors are active when PAP levels are elevated.

We then analyzed the expression of ABA signaling genes acting in the molecular cascade downstream of PYR1/PYLs/RCARs (*Hauser et al., 2011*) to gain insights into possible candidate genes that complemented *ost1*-2 *sal1*-8. While some PP2Cs were differentially expressed in *ost1*-2 *sal1*-8 compared to wild type and *ost1*-2 (gene cluster I, *Figure 3—figure supplement 2C*), none of the changes could readily explain the complementation in response to ABA; for example, expression of *ABI1* was unchanged.

There are three SnRK2 kinases (including OST1) involved in ABA signaling, but their expression did not substantially change in any genotype under both control and ABA treatments (*Figure 3—figure supplement 2D*). Since a small degree of ABA signaling is still present in *ost1* but is completely abolished in the *ost1* (*snrk2.6*) *snrk2.2 snrk2.3* triple mutant (*Fujii and Zhu, 2009*; *Nakashima et al., 2009*), we investigated whether PAP signaling requires SnRKs by generating a quadruple Col-0 mutant that lacks functional OST1 (SnRK2.6), SnRK2.2, SnRK2.3 and SAL1 (*sal1*-6). The triple *snrk2.2 snrk2.3 snrk2.6* mutant had severely impacted viability and fecundity, and wilted under ambient humidity as expected [*Figure 5B* and (*Fujii and Zhu, 2009*)]. Significantly, loss of SAL1 restored turgor in the absence of all SnRKs (*Figure 5B*), which would suggest the other SnRK2s are not compensating for the loss of OST1 in *ost1*-2 *sal1*-8.

Interestingly, the aforementioned set of characterized and putative ABA signaling genes up-regulated in *ost1*-2 *sal1*-8 contain 21 genes that are either ABA-inducible in wild type but constitutively up-regulated in *ost1*-2 *sal1*-8; or not transcriptionally ABA-inducible in wild type leaves but ABA-inducible in *ost1*-2 *sal1*-8 (*Supplementary file 2*). The gene list includes transcription factors, CDPKs and CIPKs which have the potential to individually or collectively function to restore ABA sensitivity in *ost1*-2 *sal1*-8, particularly since many of these proteins have already been shown to be key regulators of ABA responses (see Discussion). Significantly, many of the CDPKs in this list are related to group II CDPKs known to regulate SLAC1, but their function(s) remain unverified. Therefore, we investigated whether PAP could also regulate novel ABA signaling proteins in addition to the known proteins.

As an exemplar proof-of-concept, we investigated four CDPKs, CDPK32, 34, CRK2 and 8, which are largely uncharacterized with respect to their functions in guard cells despite their close sequence relatedness to other subgroup II CDPKs known to regulate SLAC1 (*Cheng et al., 2002*; *Geiger et al., 2010*; *Brandt et al., 2012, 2015*). The four CPDKs were up-regulated in the leaves of *ost1*-2 *sal1*-8, as evidenced in both transcriptomes and qPCR experiments (*Figure 5E*). Significantly, all four CDPKs were also up-regulated in *ost1*-2 xrn2-1 *xrn3*-3, suggesting that their transcription can be regulated by PAP-XRN signaling (*Figure 5E*). We undertook a series of biochemical and electrophysiological approaches since CDPKs are typically functionally redundant (*Boudsocq and Sheen, 2013*) and sextuple mutants and overexpression may have confounding effects. Since a key

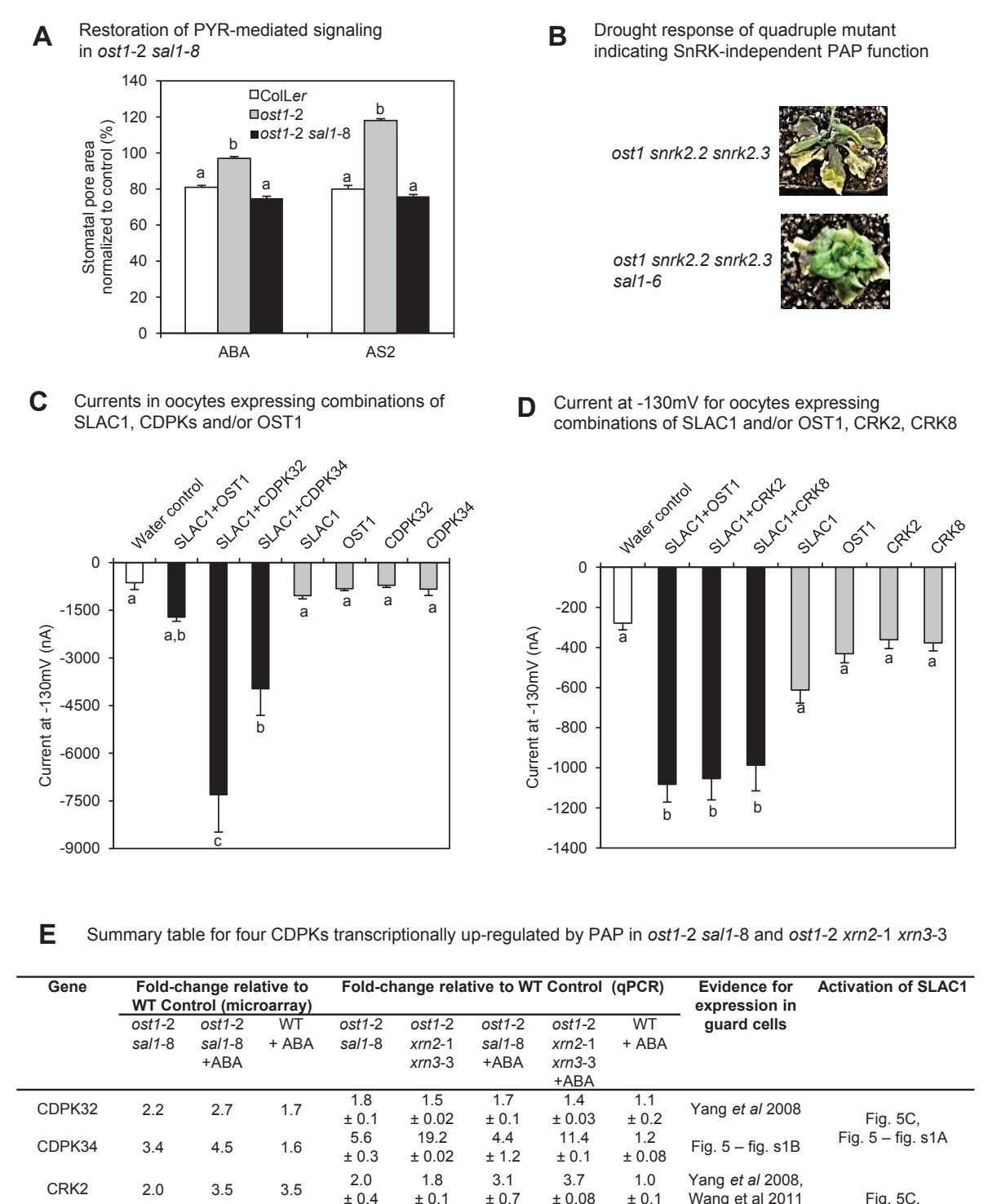

**Figure 5.** Identification of a mechanism by which PAP restores ABA sensitivity in *ost1-2 sal1-8*. (**A**) Restoration of ABA signaling mediated by specific PYR/PYLs in *ost1-2 sal1-8*. Stomatal closure was observed in leaf peels after 30 min exposure to ABA or the ABA limited-spectrum agonist, AS2. Values are mean ± SEM of 17–85 stomata per genotype per treatment. Statistically significant differences in response to a treatment are indicated (a, b, c). (**B**) Restoration of turgor to the SnRK2 gene family triple knockout ost1(*snrk2.6*) *snrk2.2 snrk2.3* by *sal1-6*. (**C**) and (**D**) Anion channel activity, expressed as
*Figure 5 continued on next page*

*Figure 5 continued*

steady state currents activated at −130 mV, in *Xenopus laevis* oocytes expressing (**C**) SLAC1, OST1, CDPK32, CDPK34 alone, or SLAC1 in combination with one of the kinases; and (**D**) SLAC1, OST1, CRK2, CRK8 alone, or SLAC1 in combination with one of the kinases. Values are means of three to five oocytes ± SEM. Significant differences between treatment (ANOVA, $p < 0.05$) are denoted a, b, c. (**E**) Summary information for four CDPKs up-regulated by PAP in *ost1-2 sal1-8*, and in *ost1-2 xrn2-1 xrn3-3*. Means ± SEM are shown for qPCR performed on three biological replicates per genotype.

The following figure supplement is available for figure 5:

**Figure supplement 1.** Expression and activity of CDPKs up-regulated by PAP.

role for CDPKs is phosphorylation of the anion channel SLAC1 (*Geiger et al., 2010*; *Brandt et al., 2012*, *2015*), and ion channel activity is restored in *ost1-2 sal1-8* (*Figure 3*), we tested the ability of the four CDPKs to regulate SLAC1 activity. In *Xenopus laevis* oocytes, strong and significant activation of SLAC1 anion currents were elicited upon co-expression of CDPK32 or CDPK34 with SLAC1 compared to non-kinase injected or water injected controls (*Figure 5C*, *Figure 5—figure supplement 1A*). CRK2 and CRK8 also activated SLAC1 comparable to that by OST1 (*Figure 5D*, *Figure 5—figure supplement 1A*). Consistent with these results, *CDPK34* is expressed in *ost1-2 sal1-8* guard cells (*Figure 5—figure supplement 1B*) and guard cell expression is already reported for *CDPK32*, *CRK2* and *CRK8* (*Yang et al., 2008*; *Wang et al., 2011*; *Boudsocq and Sheen, 2013*).

Given that ABA induces cytosolic $Ca^{2+}$ ($[Ca^{2+}]_{cyt}$) transients which activate the CDPKs and CBLs/CIPKs known to regulate stomatal closure (*Batistic and Kudla, 2009*; *Geiger et al., 2010*; *Brandt et al., 2012*; *Hubbard et al., 2012*; *Brandt et al., 2015*) we measured $[Ca^{2+}]_{cyt}$ transients in 29 different yellow cameleon (YC3.6) guard cells of leaf peels treated with PAP. During our measurements we observed cells with spontaneous $[Ca^{2+}]_{cyt}$ increases and cells without $[Ca^{2+}]_{cyt}$ transients, which were consistent with previous observations (*Grabov and Blatt, 1998*; *Staxen et al., 1999*). These fluctuations clustered into three types of oscillation patterns, but significantly in none of these were $[Ca^{2+}]_{cyt}$ transients increased by PAP (*Figure 6*).

Since PAP up-regulated multiple $Ca^{2+}$ signaling proteins but does not influence calcium transients, we investigated whether PAP instead requires $Ca^{2+}$. Indeed, PAP-mediated closure relied on a sufficient concentration of $Ca^{2+}$ (*Figure 7A*) and was prevented by addition of intracellular and extracellular $Ca^{2+}$ chelators (*Figure 7B*). Also, the kinetics and extent of PAP-mediated closure in leaf peels were attenuated compared to ABA by low exogenous calcium in the treatment buffer (*Figure 7C*). The enhanced ability of ABA compared to PAP in closing stomata under low calcium reflects ABA-mediated release of internal $Ca^{2+}$ stores and activation of $Ca^2$-independent SnRKs.

In *Figures 3* and *4* we observed that ROS bursts in response to ABA are restored in *ost1-2 sal1-8* and *ost1-2 xrn2-1 xrn3-3*; and exogenous PAP induces ROS bursts in *ost1-2*. Therefore we tested the dependency of PAP-mediated stomatal closure on ROS production by NADPH oxidases, which are $Ca^{2+}$-regulated. Inhibiting ROS production by NADPH oxidases using the NADPH oxidase inhibitor diphenyleneiodonium (DPI) inhibited PAP-mediated stomatal closure (*Figure 7D*). Collectively the results suggest that PAP-XRN retrograde communication can participate in ABA signaling and stomatal closure through an interaction with $Ca^{2+}$.

## The interaction between PAP and ABA signaling extends beyond guard cells and stomatal closure to seed dormancy and germination

To further test the general applicability of our observations on the intersection of PAP with ABA and guard cell signaling we moved to another important biological process that is distinct physiologically, but still preserves many of the components of ABA signaling in guard cells, namely seed dormancy and germination. We quantified PAP in dry seeds of *sal1-6* and it is indeed elevated compared to wild type (*Figure 8A*).

We investigated whether the elevated PAP content in *sal1* seeds affected seed dormancy and germination. In freshly-harvested seeds, *sal1-6* had increased dormancy compared to wild type (*Figure 8B*). Furthermore, *sal1-6* seeds that were after-ripened at room temperature to break seed dormancy still germinated slower than similarly treated wild type seeds (*Figure 8C*).

Next, we investigated if there is an interaction between genetically-elevated PAP and ABA in seeds. We found that *sal1* seed had enhanced repression of germination in response to exogenous

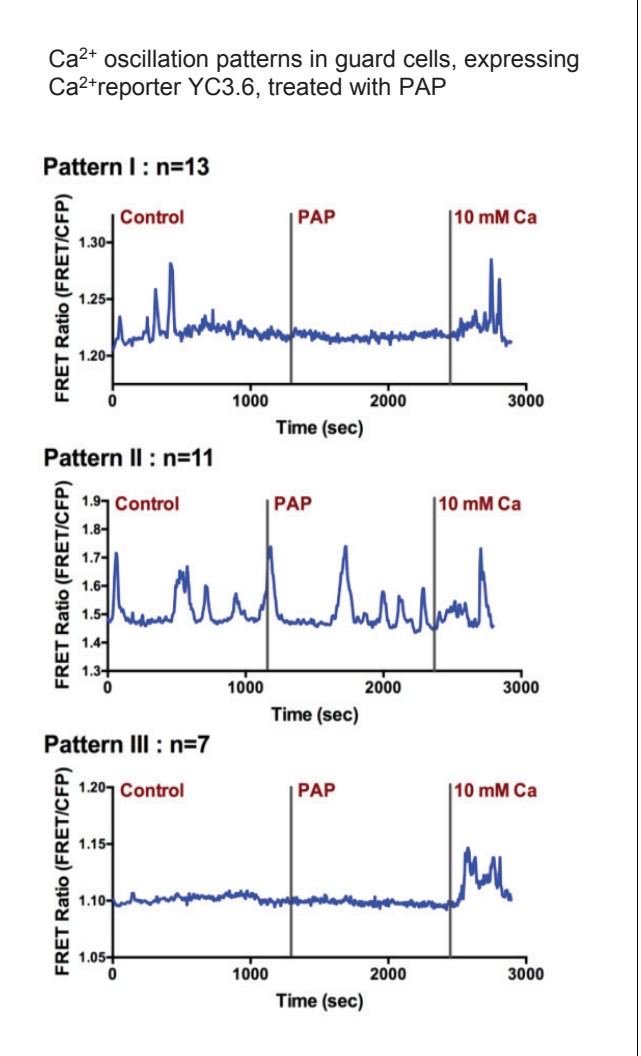

**Figure 6.** Exogenous PAP does not stimulate cytosolic $Ca^{2+}$ transients in guard cells. Three representative $Ca^{2+}$ oscillation patterns obtained from time-resolved $Ca^{2+}$ imaging experiments with PAP treatment. Numbers of observed cells in each group are labeled above each graph (measurements were obtained from 10 different plants). Guard cells of YC3.6 plants were monitored for their FRET emission at 535 nm and 480 nm. FRET ratio increases after the addition of 10 mM $Ca^{2+}$ suggesting the cells are capable of sensing intracellular $Ca^{2+}$ level changes.

ABA and paclobutrazol (a gibberellic acid biosynthesis inhibitor) and was most pronounced when both ABA and paclobutrazol were added (*Figure 8D*).

To further assess the interaction of PAP with ABA and GA on seed germination, we investigated the responses of wild type, *abi1*-1 and *abi1*-1 *sal1*-8 seed [*abi1*-1 seeds are also ABA insensitive (*Koornneef et al., 1984*), whereas *ost1*-2 does not have a seed phenotype]. As expected exogenous ABA and paclobutrazol had no significant effect on *abi1*-1 germination rates, but ABA responsiveness was restored in *abi1*-1 *sal1*-8, albeit not to the same extent as wild type (*Figure 8E*).

We then undertook a twice-replicated experiment using combinatorial treatments of ABA (0, 1, 2 µM), paclobutrazol (0.5 µM) and PAP (10, 50, 100, 500 and 1000 µM) on intact wild type and *abi1*-1 seeds. The combined results are plotted as a heat map with the corresponding analysis of variance (*Figure 8F*), and, a subset of the experiment is plotted (*Figure 8G*). PAP treatment significantly repressed wild type and *abi1*-1 germination in combination with ABA and paclobutrazol in a dose-dependent manner, but not by itself. The combinatorial treatments lowered Col-0 germination from

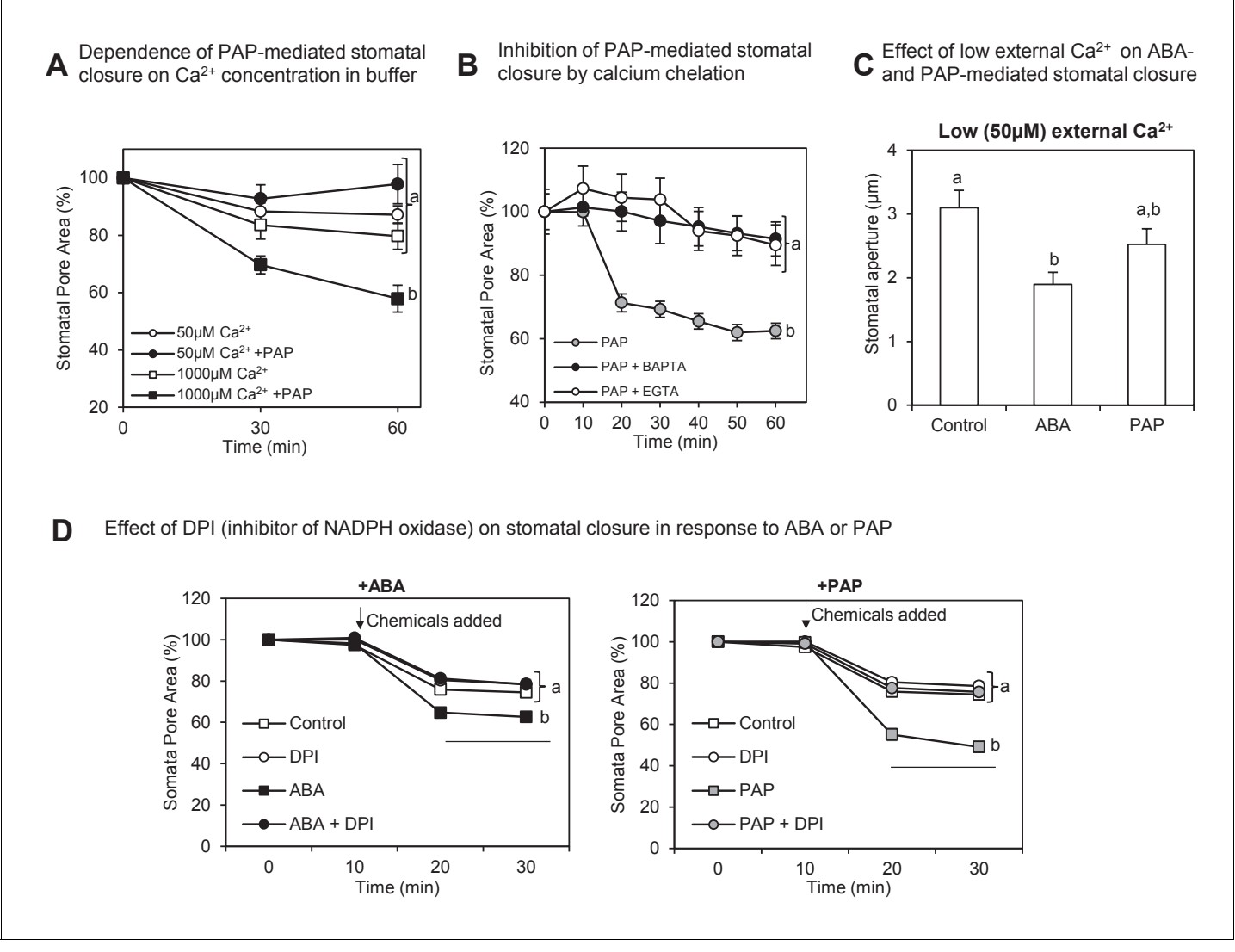

**Figure 7.** Interaction between PAP-mediated signaling with $Ca^{2+}$. (**A**) Wild type stomatal aperture with or without PAP in the presence of low (50 μM) $Ca^{2+}$ or high (1 mM) $Ca^{2+}$ in the measuring buffer. Means ± SEM for 9–10 stomata per treatment are shown. Significant differences between treatments at t = 60 min (ANOVA, $p<0.05$) are indicated by a, b. (**B**) Stomatal aperture in leaf peels treated with PAP and an intracellular calcium chelator (BAPTA-AM) or an extracellular calcium chelator (EGTA).Values are relative to control (measuring buffer). Means ± SEM for 18 stomata from four plants per treatment are shown. Significant differences between treatments at t = 60 min (ANOVA, $p<0.05$) are indicated by a, b. (**C**) Stomatal aperture in leaf peels treated with control (ethanol), 100 μM PAP or 10 μM ABA in the presence of low $Ca^{2+}$ (50 μM). Means and SEM for four plants with >28 stomata per plant are shown. Significant differences between treatments (ANOVA, $p<0.05$) are indicated by a, b. (**D**) Stomatal aperture in leaf peels that were pretreated with 20 μM diphenyleneiodonium (DPI) prior to treatment with either 100 μM PAP or 100 μM ABA, in measuring buffer. The control treatment was leaf peels which were not pretreated with DPI and were treated with measuring buffer. Values are means ± SEM for 41–51 stomata per treatment. Final level of closure was considered by ANOVA after treatment; significant difference ($p<0.05$) denoted a, b.

100% to <20% and for *abi1*-1 from 100% in the presence of 1 or 2 μM ABA to <50%. Thus, the genetic and biochemical evidence indicate that PAP functions in ABA-mediated germination control and complements *abi1*-1 in a manner analogous to results in guard cells.

Of interest was the observation that PAP content in wild type seeds was seven-fold higher than in leaves (*Figure 8A*). In leaves, oxidatively-stressed chloroplasts inhibit SAL1 activity but do not alter its abundance (*Chan et al., 2016b*); whereas during seed desiccation the chloroplasts in seeds de-differentiate into small non-photosynthetic plastids (*Mansfield et al., 1991*; *Mansfield and Briarty, 1992*; *Liebers et al., 2017*). Therefore we investigated whether PAP accumulation in seeds similarly

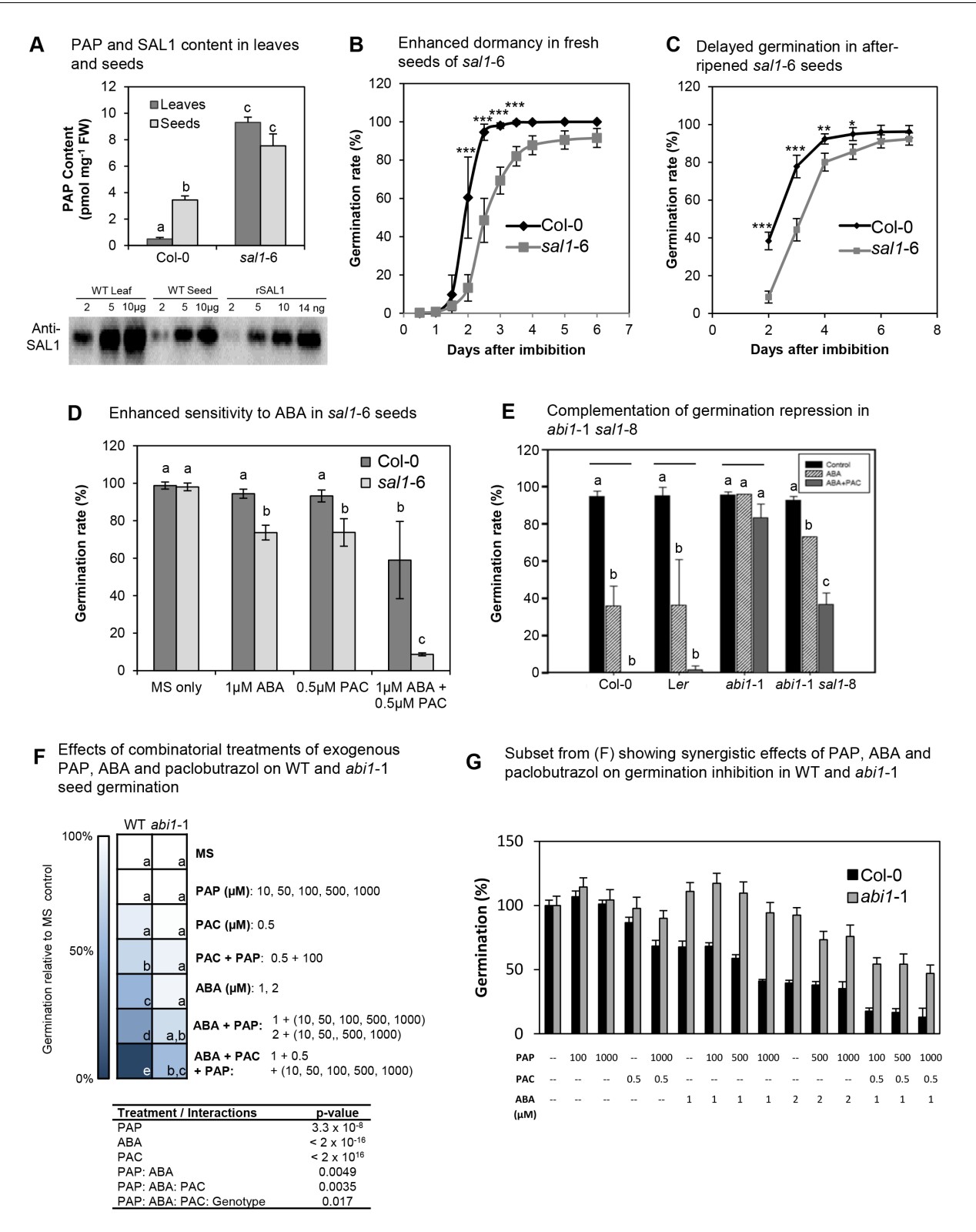

**Figure 8.** PAP also acts as an ABA secondary messenger in Arabidopsis seeds. (**A**) PAP levels in wild type and *sal1*-6 leaves and seeds. Data are means ± SEM, n = 3. Significantly different groups (p<0.05) are shown as a, b, c. Western blot of total protein probed with anti-SAL1, together with rSAL1 recombinant protein control. (**B**) Germination rates (radicle emergence) of wild type and *sal1*-6 seeds harvested fresh and plated straight onto 0.8% agarose without prior storage, stratification or sterilization. Values show averages of five to eight plates containing at least 70 seeds per plate ± SEM. *Figure 8 continued on next page*

*Figure 8 continued*

Significant differences at p<0.001 (\*\*\*) or at p<0.05 (\*) compared to wild type are shown. (C) Germination rates (radicle emergence) of wild type and *sal1*-6 non-stratified seeds that were after-ripened at room temperature for one month post-harvest. Values show averages of five plates containing at least 70 seeds per plate ± SEM. Significant differences at p<0.001 (\*\*\*), p<0.01 (\*\*) or p<0.05 (\*) compared to wild type are shown. (D) Germination rates of mature dry wild type and *sal1*-6 seeds sterilised and sown on MS plates with appropriate chemicals added before stratification and growth. At least 70 seeds were sown in a plate per genotype and eight replicates per genotype per treatment were prepared. Data are means ± SD. Significant difference groups (p<0.05) are shown (a, b, c). (E) Germination rates of wild type, *abi1*-1 and *abi1*-1 *sal1*-8 in the presence of 1 μM ABA or ABA + 0.5 μM of a GA biosynthesis inhibitor: paclobutrazol (PAC). Bar graphs represent the average of two independent experiments (n = 90 seeds per genotype per experiment) ± SEM. Significant difference (p<0.05, \*) compared to respective controls. n.s: not significant. (F) Germination rates of WT and *abi1*-1 seeds under combinatorial treatments of ABA (1, 2 μM), paclobutrazol (0.5 μM) and PAP (10, 50, 100, 500, 1000 μM), shown as a heat map. Multivariate analysis of the complete dataset shows highly significant treatment interactions (table) and were used to calculate significant difference groups (a, b, c, d, e; p<0.05) within a genotype. (G) Subset result of (F) showing individual germination rates for WT and *abi1*-1 in response to specific combinations of ABA, paclobutrazol and PAP. For both (F) and (G) n = 70 seeds x 5 plates = 350 seeds per genotype per combinatorial treatment.

The following figure supplement is available for figure 8:

**Figure supplement 1.** Role of SAL1 and PAP in regulating seed germination.

reflected the oxidizing environment of seed desiccation, or, other mechanisms such as regulation of SAL1 abundance. *SAL1* transcript abundance in seeds is the lowest amongst all *Arabidopsis* developmental stages and dry seed imbibition increases SAL1 expression greater than 10-fold, constituting the most dramatic change in SAL1 expression across 3000 different perturbations (*Figure 8—figure supplement 1A,B*) (*Zimmermann et al., 2004*). Associated with the elevated PAP and lower *SAL1* mRNA levels we did observe substantially decreased SAL1 protein abundance in seeds compared to leaves (*Figure 8A*). Therefore unlike leaves, in seeds PAP accumulation and signaling may be modulated by a downregulation of SAL1 abundance.

## Discussion

In this study we present evidence for organelle retrograde signaling functioning in concert with hormonal signaling in guard cells to mediate stomatal closure and in seeds to mediate dormancy and germination. This new role was discovered *via* the unexpected ability of PAP to restore ABA-responsiveness in the two canonical *ost1*-2 and *abi1*-1 mutants that function near the apex of ABA signaling and further demonstrated independently in wild type plants (*Figures 1*, *3*, *5* and *8*, *Figure 1—figure supplement 3*). Complementation of *ost1*-2 and *abi1*–1 occurred whether PAP levels were genetically or exogenously manipulated and traits observed to be restored include ABA-inducible gene expression, ROS bursts, ion fluxes, gas exchange and aperture. Similarly, dormancy and germination are impacted by elevated PAP in both wild type and *abi1*-1. These findings raise a series of questions. First, how do these observations relate to PAP perception and its effects on ABA biosynthesis and signaling in guard cells and seeds? Second, how can the observed effects of PAP explain the complementation of ABA-insensitive mutants? Finally, what is the physiological relevance of these findings *in planta*?

### PAP perception and its influence on ABA biosynthesis and responses

We have previously reported that ABA content can be elevated by PAP in *sal1*-8 (*Rossel et al., 2006*). Similar trends were observed herein, although the differences were not statistically significant between *ost1*-2 *sal1*-8 and *ost1*-2 due to high variability between replicates (*Figure 1—figure supplement 3*). This enables formulation of an hypothesis that the primary mechanism of action of PAP is to stimulate ABA biosynthesis. Such a function for PAP may contribute to the lower $g_s$ and seed dormancy in *sal1* and PAP-treated wild type, which have functional ABI1 and OST1 (*Figures 1* and *8*). However, would elevation of endogenous ABA restore responsiveness in mutants that are insensitive to exogenous ABA? The triple *snrk* mutant has little to no response to ABA despite accumulating three-fold more endogenous ABA than wild type (*Fujii and Zhu, 2009*; *Nakashima et al., 2009*) and similarly the *ost1*-2 mutant is largely insensitive to ABA (*Mustilli et al., 2002*; *Fujii and Zhu, 2009*; *Nakashima et al., 2009*). Yet, *ost1*-2 and the *snrk* triple mutant were complemented by *sal1*

and PAP treatments (*Figures 1*, *3* and *5*). Another expectation if PAP primarily rescued ABA sensitivity in the double mutants by enhancing ABA content is that there could be evidence for constitutive rescue and enhanced sensitivity to ABA in the double mutants in response to the elevated endogenous ABA. However, we did not observe evidence for this either in the transcriptional, stomatal aperture or $g_s$ properties of *abi1*-1 *sal1*-8 and *ost1*-2 *sal1*-8 (*Figures 1* and *3*). Therefore, while PAP may also function by up-regulating ABA levels we conclude it primarily acts by facilitating a signaling response to exogenously applied or endogenously stress-induced ABA.

For PAP to mediate ABA responses, it must be perceived by the cell. One hypothesis is that PAP could bind to, and affect, the activity of ABA signaling proteins. None of the ABA receptors have known adenosine binding domains and as the signaling downstream of these receptors is still blocked in *abi1*-1 and *ost1*-2, then direct receptor activation by PAP binding would be inconsistent with the restoration of ABA perception in the double or quadruple *snrk sal1* mutants (*Figures 1B* and *4B*) and *abi1*-1 *sal1*-8 (*Figures 1* and *8*). PAP has been reported to also bind to poly (ADP-Ribose) Polymerase (PARP) and Nucleotide Diphosphate Kinase (NPDK) proteins in mammals (*Schneider et al., 1998*; *Toledano et al., 2012*), but in plants no interaction partners other than XRNs have been described for PAP. Nor is PAP known to bind any ABA signaling proteins. Although additional PAP targets cannot be precluded our results herein indicate that PAP is primarily perceived and participates in ABA signaling through its established retrograde signaling target proteins, the XRNs (*Figure 4*, discussed below).

In retrograde signaling PAP levels are regulated in the chloroplast (*Chan et al., 2016b*), it is transported into the cytosol (*Gigolashvili et al., 2012*) and its only reported sites of perception in plants are nuclear exoribonucleases (*Estavillo et al., 2011*; *Hirsch et al., 2011*). Herein we observe that *ost1*-2 *xrn2*-1 *xrn3*-3 phenocopies *ost1*-2 *sal1*-8 at multiple levels of ABA responses and exhibits restoration of ABA-dependent stomatal closure in whole plants and leaf peels, ROS burst and ion fluxes (*Figure 4*). As further discussed below, *ost1*-2 *xrn2*-1 *xrn3*-3 also phenocopies *ost1*-2 *sal1*-8 in upregulating expression of CDPKs capable of regulating SLAC1 activity in oocytes. Additionally, the petiole feeding experiments where we co-fed ATP (which we propose would inhibit import of PAP into the chloroplast) or LiCl [which is reported to inhibit chloroplastic SAL1 (*Quintero et al., 1996*)] provide results consistent with modulation of the intracellular PAP pathway. Therefore, while we do not exclude the possibility that PAP is partially sensed inside or outside the cell in an XRN-independent manner; given the similarity of complementation whether endogenously manipulated or exogenously applied to seeds, leaves and epidermal peels we favor the explanation that PAP acts as an agonist of ABA physiological responses by acting *via* XRNs in its established retrograde signaling pathway.

The exogenous PAP-mediated closure within 10–30 min (*Figure 2*) could be consistent with a transcriptional response, as nuclear transcription in response to chloroplast stress can commence within 20–60 s (*Vogel et al., 2014*; *Suzuki et al., 2015*). Eukaryotic transcription and translation rates are also sufficiently rapid as only three minutes are required for the de novo synthesis of a typical 50 kDa protein (*Milo and Phillips, 2015*).

A large proportion of the genes induced by ABA in wild type leaves were not induced in *ost1*-2, but were significantly induced in *ost1*-2 *sal1*-8 leaves, such that they were no longer significantly different to wild type plus ABA, albeit attenuated. Additionally, various ABA signaling genes, including multiple $Ca^{2+}$ sensor proteins (CDPKs, CBLs, CIPKs) that regulate ion channel activity either directly by phosphorylation or indirectly *via* ROS production (*Batistic and Kudla, 2009*; *Simeunovic et al., 2016*), as well as two transcription factors that each regulate multiple genes, were up-regulated in *ost1*-2 *sal1*-8 +/− ABA (*Supplementary file 2*). The transcriptional response in *ost1*-2 *sal1*-8 +/− ABA reports on two facets of PAP- ABA interactions: one is that ABA-signaling is indeed restored at the transcriptional level as well as the physiological level; the second is that it may provide insight into which proteins are key for this restoration.

## How does PAP interact with ABA and guard cell signaling?

Genes simply restored to wild type levels after ABA treatment in *ost1*-2 *sal1*-8 more likely reflect the effect of complementation, not the cause of it. If the mechanism of PAP complementation is at least in part transcriptionally regulated then candidate gene(s) would need to collectively restore ABA responsiveness and be constitutively up-regulated in *ost1*-2 *sal1*-8 by PAP. The second category would be genes that are non-ABA inducible in wild type leaves but are ABA-inducible in *ost1*-2 *sal1*-

8 (*Supplementary file 2*). This for example could include ABA receptors, PP2Cs or SnRKs that can bypass ABI1 and OST1. However, the seven ABA receptors detected as expressed were either suppressed or unchanged and no changes in expression of PP2Cs could readily explain the complementation. Likewise, the expression of the three SnRK2 kinases did not change and the quadruple *snrk sal1*-6 mutant conferred drought tolerance arguing against a role for the other SnRKs (*Figure 5B*).

Intriguingly, PAP up-regulates 21 individual ABA / $Ca^{2+}$ signaling components that fulfill either of the requirements above (*Supplementary file 2*). The transcription factor MYB61 which is a key transcriptional activator of stomatal closure (*Liang et al., 2005*) is constitutively up-regulated in *ost1*-2 *sal1*-8 minus ABA. Additionally, MYC2 is non-ABA inducible in wild type leaves but is ABA-inducible in *ost1*-2 *sal1*-8. MYC2 regulates ABA-responsive transcription; and its overexpression leads to enhanced ABA sensitivity *via* constitutive up-regulation of ABA signaling genes (*Abe et al., 2003*).

Sixteen $Ca^{2+}$ signaling proteins (CDPKs and CBL/CIPKs) also fulfill the aforementioned criteria. For category 1, CRK2 as well as CDPKs 9, 28, 30, 32 and 34 are constitutively up-regulated in *ost1*-2 *sal1*-8. Many of these kinases have unclear functions with regards to ABA signaling, yet are related to group II CDPKs that are known to regulate SLAC1 activity. Interestingly, CDPK32 has key functions in regulating ion channels in pollen tubes (*Zhou et al., 2014*) and can also bind to and regulate the activity of the ABA transcription factor ABF4 thereby enhancing ABA sensitivity in leaves and seeds (*Choi et al., 2005*). Two CIPKs that are key regulators of ABA signaling are also constitutively up-regulated in *ost1*-2 *sal1*-8: CIPK6 activates the key guard cell $K^+$ channel AKT1 (*Lan et al., 2011*) and CIPK1 mediates ABA responsiveness in guard cells (*D'Angelo et al., 2006*). Another constitutively up-regulated gene, CIPK21, is important for stress tolerance and can interact with multiple CBLs known to regulate ABA signaling (*Pandey et al., 2015*). We also observed ABA-inducible expression of CRK8 and CDPK19 specifically in *ost1*-2 *sal1*-8 but not in wild type. Interestingly, CDPK19 regulates ROS homeostasis in guard cells (*Zou et al., 2015*). The extent to which CIPK transcriptional up-regulation can be correlated with the complementation will require careful investigation because CIPK activity is also regulated post-translationally in a complex manner through combinatorial interactions with different CBLs, PP2Cs and their target proteins (*Batistic and Kudla, 2009*).

We show that at least four additional non-canonical CDPKs, CRK2 and 8 as well as CDPK32 and 34, can activate SLAC1 in an oocyte heterologous system independently of OST1 (*Geiger et al., 2010*; *Brandt et al., 2012*, *2015*) (*Figure 5*, *Figure 5—figure supplement 1*). Two of them, CDPK32 and 34, are known to be plasma membrane localized (*Choi et al., 2005*; *Myers et al., 2009*) which is consistent with SLAC1 localization *in planta*. These results are also in line with the requirement of PAP and CDPK34 for $Ca^{2+}$ [*Figure 7* and *Myers et al. (2009)*], and increase the number of reported SLAC1 activators (*Geiger et al., 2010*; *Brandt et al., 2012*, *2015*). It would be interesting to investigate whether these CDPKs can also regulate $K^+$ channel activity, given the restoration of $K^+$ fluxes by PAP and the reported ability of other CDPKs to regulate $K^+$ channels (*Latz et al., 2013*).

So, could over-expression of one of more of the CPDKs be sufficient for complementation of ABA signaling mutants through activation of SLAC1? This hypothesis is viable for *ost1*-2 as CDPKs are not directly regulated by OST1 nor PP2Cs (*Boudsocq and Sheen, 2013*; *Brandt et al., 2015*; *Simeunovic et al., 2016*). The degree of CDPK up-regulation was strikingly similar between *ost1*-2 *sal1*-8 and *ost1*-2 *xrn2*-1 *xrn3*-3 (*Figure 5E*), and correlate with the restoration of ABA-responsive ion fluxes and stomatal closure in both genotypes. Thus, increased expression of CDPKs may be hypothetically sufficient for the complementation in *ost1*-2 *sal1*-8 and *ost1*-2 *xrn2*-1 *xrn3*-3. Indeed overexpression of CDPK32 in a wild type background has been shown to enhance ABA sensitivity in leaves (*Choi et al., 2005*). In the dominant *abi1*-1 mutant allele used in this study, however, it is unclear if phosphorylation of SLAC1 by the PAP-upregulated CDPKs alone is sufficient for complementation since there are conflicting reports on whether this dominant PP2C allele can competitively dephosphorylate SLAC1 and thereby negate CDPK action (*Geiger et al., 2009*; *Scherzer et al., 2012*; *Brandt et al., 2015*). More likely additional factors, like CIPKs 1, 6, 21, MYC2 and MYB61 are needed in combination with the CDPKs for the observed restoration of ABA perception and action in absence of OST1 or in the presence of the dominant ABI1-1 PP2C. Whether these factors function in parallel, or are part of a PAP-interaction network where up-regulation of one leads to the induction and activation of the others, requires investigation.

## Complementation of ABA-deficient and -insensitive genotypes

If PAP functions at least partly through $Ca^{2+}$ signaling proteins and requires ABA-induced $Ca^{2+}$ (*Figure 7*), how can the complementation in the various ABA biosynthesis and ABA-insensitive mutants be explained? In *aba2*-3 *sal1*-8, ABA biosynthesis is not completely inhibited, since the *aba2*-3 allele is leaky (*Léon-Kloosterziel et al., 1996*; *Laby et al., 2000*; *Barrero et al., 2005*). In *ost1*-2, ABA biosynthesis is not affected but the loss of SnRK2 activity strongly impairs ABA perception (*Mustilli et al., 2002*). Significantly, however, ABA can still activate $Ca^{2+}$ channels in the plasma membrane in guard cells of *snrk* triple mutants (*Brandt et al., 2015*), which would enable $Ca^{2+}$ entry into the cytosol. This was simulated to a degree in our epidermal leaf peels experiments. The high external $Ca^{2+}$ in the measuring buffer diffuses into the cytosol where it activates $Ca^{2+}$ transients (*Figure 6*) and enabled exogenous PAP to close *ost1*-2 stomata. This suggests that entry of $Ca^{2+}$ into the cytosol and its intracellular oscillations in response to drought and/or ABA may be sufficient to activate the PAP- and XRN-upregulated proteins, thus restoring ABA sensitivity and stomatal closure in *ost1*-2 *sal1*-8 and *ost1*-2 *xrn2*-1 *xrn3*-3 (*Figures 1B,C* and *4*).

In *abi1*-1, however, the mutated PP2C protein is insensitive to ABA/PYR-mediated inhibition and presumably there is stronger antagonism of $Ca^{2+}$ signaling proteins by PP2Cs in this mutant. Indeed, we observed that PAP accumulation in *abi1*-1 *sal1*-8 only partially restored germination inhibition by exogenous ABA and paclobutrazol (*Figure 8F*). Similarly, the germination inhibition of *abi1*-1 by combinatorial treatments of exogenous PAP, ABA and paclobutrazol occurred to a lesser extent compared to wild type; and the synergistic interaction between exogenous PAP and ABA required a higher ABA concentration in *abi1*-1 (*Figure 8G*). Regardless, our results in guard cells and seeds collectively indicate that PAP is indeed able to act in ABA signaling and partially complement *abi1*-1. It may be ABA-mediated inhibition of the remaining eight group A PP2Cs (*Hauser et al., 2011*; *Komatsu et al., 2013*) in conjunction with PAP-XRN-mediated upregulation of transcription factors, CDPKs and CIPKs (*Supplementary file 2*) can alter the transcription of downstream ABA signaling genes as well as the ratio between phosphorylation and dephosphorylation of multiple target proteins. Future research would seek to determine the exact nature of the interrelationship between ABA, $Ca^{2+}$ and PAP.

## Physiological functions and a model for PAP in ABA signaling

Our findings raise the question of why and for what purpose do plant cells utilize PAP to mediate ABA responses, if the canonical pathway mediated by ABI1 and OST1 is sufficient to close stomata and regulate seed germination. Exogenous and endogenous manipulation of PAP in wild type and *sal1*-8 genotypes where ABI1 and OST1 are still functional lowers stomatal conductance and raises leaf temperature under constitutive conditions and generates a hypersensitive response to exogenous ABA (*Figures 1* and *2*), and this is replicated in seeds where PAP enhanced germination inhibition by ABA (*Figure 8*). Collectively, our results indicate that ABA and PAP may act synergistically.

We propose the following working model of how PAP may facilitate ABA-mediated regulation of stomatal function under physiological conditions, thereby integrating cellular responses to oxidative stress and tuning hormonal pathways (*Figure 9*). Oxidative stress during drought alters chloroplast redox poise to induce a 30-fold PAP accumulation *via* redox regulation of SAL1 (*Estavillo et al., 2011*; *Chan et al., 2016b*). PAP accumulation could activate its downstream signaling through XRNs to transcriptionally up-regulate multiple ABA/$Ca^{2+}$ signaling proteins that are normally lowly expressed, and this includes four CDPKs that can activate SLAC1 in oocytes. These observations provide mechanisms for the observed bypass of the canonical pathway in *abi1*-1 *sal1*-8, *ost1*-2 *sal1*-8, and *ost1*-2 *xrn2*-1 *xrn3*-3; and, stomatal closure in PAP-treated leaf peels in the presence of $Ca^{2+}$ as well as the ABA hypersensitivity in *sal1*-8. Interestingly, combinatorial stresses are known to activate additional transcriptional responses compared to single stresses (*Suzuki et al., 2014*). Our previous findings also showed that *sal1*-8 with higher PAP content has more closed stomata relative to wild type under high light (*Rossel et al., 2006*). This raises an open question as to whether combinatorial chloroplastic oxidative stresses, such as drought and high light, might lead to more rapid or enhanced PAP signaling to regulate stomata in conjunction with ABA. Interestingly, the role of PAP in ABA signaling in mature seeds suggests that PAP could also be a signal from small, non-photosynthetic immature plastids which de-differentiated from chloroplasts during seed desiccation (*Mansfield et al., 1991*; *Mansfield and Briarty, 1992*; *Liebers et al., 2017*). The regulation of PAP-

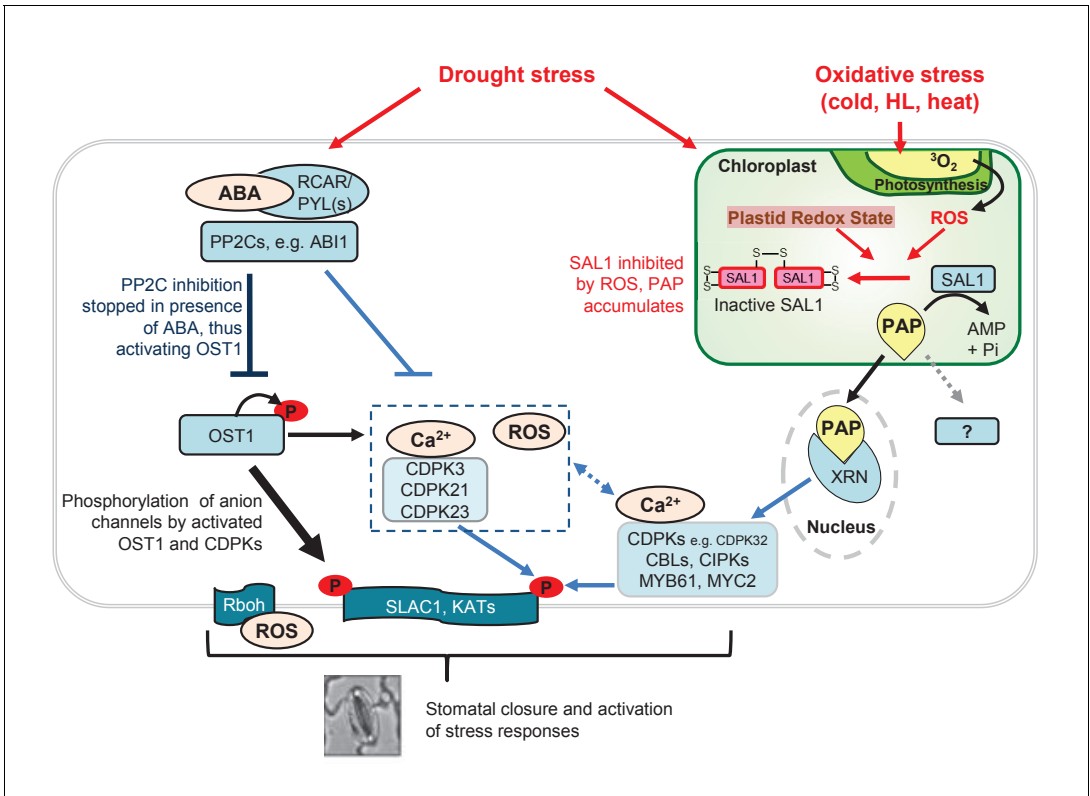

**Figure 9.** Model for fine-tuning of stomatal closure by PAP retrograde signaling. Proposed intersection between PAP and ABA signaling during drought stress, or in *ost1-2 sal1*-8 / *abi1*-1 *sal1*-8 treated with ABA. Binding of ABA to its receptors (PYR/PYLs) inactivates the inhibitory PP2Cs, thus allowing activation of OST1 for phosphorylation of proteins such as transcription factors and SLAC1. This is the major pathway for stomatal closure. Additionally, ABA signaling results in $Ca^{2+}$ release which could activate multiple $Ca^{2+}$ signaling proteins including CDPKs and CBLs/CIPKs thus allowing phosphorylation and activation of SLAC1. Many of these ABA and $Ca^{2+}$ signaling proteins can be regulated by PAP predominantly via PAP-XRN-mediated retrograde signaling. The CDPK activation of SLAC1 can occur in parallel with $Ca^{2+}$-independent OST1, allowing a convergence between chloroplast and ABA signaling at this key anion channel for stomatal closure. It is possible that PAP can also regulate stomatal closure through other proteins, though there is as yet no evidence for this in plants. The relative contributions of each pathway towards control of stomatal closure are indicated by the thickness of the arrows and lines. Solid lines and arrows indicate characterized pathways. Signaling pathways which have not been fully studied are indicated with dashed lines and '?'.

ABA interaction might occur differently in seeds compared to guard cells, since PAP accumulation appears to be regulated more by SAL1 abundance than chloroplast redox in seeds.

The canonical ABA pathway mediated by PYL, ABI1, OST1 and $Ca^{2+}$ transients regulate multiple processes in various tissues and specialized cells, including stomatal closure and aspects of seed germination. Our findings provide unanticipated insights into how guard cell or mesophyll chloroplasts may have input into guard cell regulation by mediating an additional ABA pathway that is complementary to and independent of the OST1 signaling network. The ability of SAL1-PAP-XRN retrograde signaling to intersect with ABA signaling in multiple tissues and its complex interactions with $Ca^{2+}$ and ROS raise the hypothesis that PAP has attributes of a secondary messenger. Consequently, PAP-mediated retrograde signaling can now be viewed as more than bilateral communication between organelles and the nucleus as it can act cooperatively with hormone signaling and influence broader physiological outcomes in specialized cells and multiple tissues.

# Materials and methods

## Plant material, growth and stress treatments

Seeds were germinated in soil and kept at 4°C for three days to synchronize germination. Seedlings were grown at 100–150 µmol photons m$^{-2}$ s$^{-1}$, 12 hr photoperiod, 21–23°C and 50–55% humidity, unless otherwise stated. The *alx8* (*sal1*-8, Col-0 background) mutant was crossed with *abi1*-1 (*Koornneef et al., 1984*), *ost1*-2 (*Mustilli et al., 2002*) in the L*er* background in order to generate double mutants. Homozygous F2 plants were screened using derived cleavable amplified polymorphic sequence (dCAPS) markers to confirm the presence of individual mutations and sequenced. The Col-0 L*er* F1 hybrid was generated as a control and in most experiments wild type refers to the F1 hybrid; otherwise both parental genotypes were used. The *ost1*-2 *snrk2.2 snrk2.3* triple mutant was obtained from Prof. Sean Cutler (University of California, Riverside) and crossed to the SAL1 null allele *fry1*-6 (sal1-*6*) to generate the quadruple mutant. The *cas* mutant was obtained from Prof. Marc Knight (Durham University). Drought stress treatment was performed as previously reported (*Wilson et al., 2009*; *Estavillo et al., 2011*).

ABA at a concentration of 20 µM in water was root fed to hydroponically grown plants or sprayed onto leaves using a Studio Series IS-875 with a 0.5 mm nozzle opening (Iwata). For the investigation of the effect of different chemicals on stomatal closure in whole leaves via petioles, 20 µM ABA (Sigma-Aldrich), 100 mM LiCl, or 1 mM ATP, 1 mM or 100 µM PAP (Sigma-Aldrich) alone or in different combinations were prepared in infiltration buffer (1 mM PIPES KOH pH 6.0, 1 mM Sodium citrate, 1 mM KCl, 15 mM Sucrose) modified from a cordycepin infiltration buffer (*Seeley et al., 1992*).

For epidermal peels, plants were grown under approximately 80% RH humidity and 12 hr/12 hr and 22/20°C light/dark cycle in a growth chamber as described in *Leyman et al. (1999)* and *Eisenach et al. (2012)*.

## Protein gel electrophoresis and immunoblotting

Total protein from tissue and cell fractions was extracted in 10% (w/v) tricarboxylic acid in cold acetone. Western blots were performed as previously described (*Wilson et al., 2009*). In brief, 5 µg of leaf total protein extract and 5 ng of recombinant SAL1 (rSAL1) used as a positive antibody specificity control were resolved on 4–12% (w/v) SDS-PAGE (NuPAGE, Invitrogen), electro transferred to a nitrocellulose membrane and probed with a 1:1000 dilution of polyclonal antibodies raised against rSAL1 (Research Resource Identifier AB_2183225) (*Wilson et al., 2009*) for 10 min. After three washes with PBS, the blot was incubated with 1:10000 dilution of HRP-conjugated goat anti-rabbit IgG for 10 min, washed three times and developed using the Super Signal West FemtoChemiluminescent detection kit (Pierce) for 5 min.

## Quantification of metabolites

Total adenosines were extracted with 0.1 M HCl, derivatization with chloroacetaldehyde and quantified fluorometrically after HPLC fractionation as previously described (*Estavillo et al., 2011*). 3′-phosphoadenosine 5′-phosphate (PAP) quantification was performed by integrating the HPLC peak area and converting these to pmol units using standard curves of 1, 5 and 10 pmol standard (*Bürstenbinder et al., 2007*).

Total leaf ABA content was quantified using a modified ELISA-based method (*Van Norman et al., 2014*). Approximately 100 mg of leaf tissue was harvested, immediately flash frozen with liquid nitrogen, and ground to a fine powder with a 1/8′ steel ball bearing in a 2 ml Eppendorf tube at 25 Hz for 2 min using the Tissue Lyser II (Qiagen, Germany). ABA was extracted from the ground tissue by shaking overnight (25 rpm, 4°C) in 2 ml of 80% (v/v) methanol, followed by centrifugation (16,000 g) at 4°C. The supernatant was collected, dried under vacuum to 25% of its original volume, and acidified to pH 3.0 using 0.5 M HCl. An equal volume of ethyl acetate was then added and allowed to mix by gentle inversion for 10s. After standing for 30 s the upper organic layer of ethyl acetate was transferred to another 2 ml Eppendorf tube and dried under vacuum to complete dryness. The resulting pellet was resuspended in 1 ml of 1× Tris- Buffered Saline (TBS) [3.03 g/L Trizma base, 5.84 g/L Sodium chloride, 0.2 g/L Magnesium chloride hexahydrate, 0.2 g/L Sodium azide, pH 7.4]. A 1:10 dilution in TBS of each sample was used for the quantification assay, which was

performed using the Phytodetek ABA Quantification Kit (Agdia, USA) according to the manufacturer's instructions.

## Gas exchange measurements

Six-week-old hydroponically grown plants were transferred from nutrient solution (*Gibeaut et al., 1997*) to the nutrient solution containing 20 µM ABA (Sigma-Aldrich). The effect of ABA on stomatal conductance ($g_s$) was measured after 2 hr of incubation (three leaves per plant) using an AP4 porometer (Delta-T Devices, UK). For *Figure 1—figure supplement 2*, measurements of *abi1*-1 and *abi1*-1 *sal1*-8 were performed separately to the other genotypes.

## Infra-red thermography

Leaf temperature was monitored over time using an infra-red (IR) camera FLIR A600-Series, IR lens f = 13.1 mm (FLIR Systems AB, Sweden). The temperature of individual leaves was monitored using the ThermaCAMTM Researcher Pro 2.10 software (FLIR Systems AB) and tissue harvested for total RNA and adenosine extraction after 2 hr of incubation with ABA or water.

## Stomatal bioassays

Stomatal apertures in response to individual and combinatorial chemical treatments (ABA, PAP, ATP, DPI, calcium chelators, AS2) were measured in epidermal peels of newly expanded leaves of three to four-week old plants as described in *Leyman et al. (1999)* and *Eisenach et al. (2012)*. Stomatal images were taken using a bright- field microscope capable of 400× magnification for 10 min in Opening Buffer (OB: 50 mM KCl, 5 mM MES titrated to pH 6.1 with NaOH) to make sure the stomata stay open before subsequent assays in a physiological Measuring Buffer [MB: 10 mM KCl, 5 mM MES titrated to pH 6.1 with Ca(OH)$_2$ (*Blatt et al., 1990*; *Armstrong et al., 1995*); final concentration of $Ca^{2+}$ is ~1 mM] or the appropriate chemical (dissolved in MB) for another 20–50 min. The epidermal peels were under the same light intensity (150 µmol m$^{-2}$ s$^{-1}$) as in the growth chamber to avoid dark-induced stomatal closure. Stomata aperture width and length were measured using ImageJ (NIH, USA). The stomatal pore area was calculated using these values under the assumption that the area of a stomatal pore was that of an ellipse. Data are expressed as percentage over the control. For some treatments, epidermal peels were pre-incubated prior to measurement (see below).

The following chemicals were employed to test the role of $Ca^{2+}$ and ROS respectively in PAP-induced stomatal closure: intracellular $Ca^{2+}$ chelator 1,2-bis-(o-Aminophenoxy)-ethane-N,N,N',N'-tetraacetic acid AM ester (BAPTA-AM), extracellular $Ca^{2+}$ chelator ethylene glycol tetraacetic acid (EGTA), and NADPH oxidase inhibitor diphenyleneiodonium (DPI). Thirty minutes before 100 µM ABA / PAP treatment, epidermal peels were preincubated with opening buffer containing 2 mM BAPTA-AM or 20 µM DPI to ensure the diffusion of these compounds into guard cells. In a separate treatment, 5 mM EGTA was added with 100 µM PAP in the measuring buffer (MB, containing ~1 mM $Ca^{2+}$) to chelate external $Ca^{2+}$. An equivalent amount of DMSO was added to the blank MB treatments to account for the DMSO solvent used to solubilize DPI. Stomatal aperture was measured as described above.

ABA or PAP-mediated stomatal closing analyses in the presence of low $Ca^{2+}$ were performed with 3–4 week-old plants grown in a CMP3244 plant growth chamber (Conviron). Epidermal peels were prepared with a perforated Gorilla clear tape (Gorilla Glue) by a modified protocol from *Ibata et al. (2013)*. Briefly, Gorilla clear tape with a hole (approximately 1.5 × 3 mm) was attached to the abaxial side of the leaf and the same tape without a hole attached to the adaxial side. By gently pulling away two tapes, an abaxial epidermal peel was detached with Gorilla clear tape with the hole and the area inside the hole was used for stomatal movement assays. Epidermal peels were incubated in the assay buffer (5 mM KCl, 50 µM CaCl$_2$ and 10 mM MES-Tris pH 5.6) for 2 hr before treatment. ABA, PAP or EtOH as a solvent control was added to a final concentration of 10 µM or 100 µM, respectively, for 1 hr incubation. Images were acquired using an inverted light microscope (Nikon Eclipse TS100) equipped with a 40x objective and connected to the Scion camera (Scion Corporation). Initial apertures were measured before ABA or PAP treatment and the same stomates were tracked to measure apertures after treatment. Apertures were measured using Fiji (*Schindelin et al., 2012*).

## Fluorescence microscopy

The ABA- and PAP-induced accumulation of ROS in guard cells of epidermal peels was measured using 2′,7′-dichlorodifluorescein diacetate ($H_2DCFDA$, Invitrogen) according to *Wang et al. (2013)* and *Bonales-Alatorre et al. (2013)*. After 2 hr of incubation in OB, the epidermal peel was incubated with 20 µM dye dissolved in DMSO and MB for 30 min, and then rinsed with MB three times. Epidermal peels were incubated in ABA or PAP dissolved in MB (or MB only as a negative control) for 10 min. Fluorescence images were taken using a Leica inverted confocal microscope controlled by the LAS AF software (Leica Microsystems, Germany). We used the 496 nm excitation line of an argon multiline laser. $H_2DCFDA$ fluorescence emission was detected at 505 to 525 nm. Chloroplast fluorescence was detected at 680 to 700 nm in order to separate the autofluorescence of chlorophyll in guard cells. Images were quantified with ImageJ software (National Institutes of Health, USA) according to *Burgess et al. (2010)*. The background signal was measured from an empty region of a similar size and subtracted from the stomata signal to obtain corrected total stomatal fluorescence values. Each stomate was considered to represent a single biological unit and thus fluorescence values were averaged across all stomata at each time point for each treatment. At least three biological replicates were measured for each treatment.

## Guard cell ion flux measurements

The preparation of epidermal peels for ion flux measurements was identical to the stomatal bioassay. Net fluxes of $K^+$ and $Cl^-$ were measured noninvasively using ion-selective vibrating microelectrodes (the MIFE technique) essentially as described in *Chen et al. (2005)*. Epidermal peels were pretreated with stomatal Opening Buffer for 2 hr (*Armstrong et al., 1995*) before ABA, PAP and $CaCl_2$ treatments. Epidermal peels were fixed on coverglass coated with Dow-Corning silicon prosthetic adhesive (Factor II, Tucson, USA) and placed in a 5 ml measuring chamber containing measuring buffer (10 mM KCl, 5 mM $Ca^{2+}$-MES, pH 6.1). Four electrodes with fine tips (R = 4–6 GΩ) were filled with ion-selective cocktails (Sigma, Buchs, Switzerland) and their tips were aligned and positioned ~40 µm above the surface of the guard cells. During measurements, electrodes were moved towards and away from the sample in a slow (5 s cycle, 80 µm amplitude) square-wave by a computer-driven micromanipulator. Net ion fluxes were calculated from the measured differences in electrochemical potential for these ions between two positions. Ion fluxes were measured for 10 min in the control to ensure steady initial values before adding 500 µM ABA and PAP and 10 mM $CaCl_2$ for another 50 min.

## Oocyte voltage clamps

Electrophysiological recordings using Xenopus laevis oocytes were described previously (*Honsbein et al., 2009*; *Grefen et al., 2010*). Constructs with SLAC1, OST1, CDPK32, CDPK34, CRK2, and CRK8 were cloned in an oocyte expression vector. Plasmids were linearized and capped cRNA was synthesized in vitro using T7 mMessage mMachine (Ambion). cRNA quality as a single band was confirmed by denaturing gel electrophoresis. cRNA was mixed to ensure equimolar ratios unless otherwise noted. SLAC1 and OST1 were co-injected in a ratio of 3:1; SLAC1 and individual CDPKs/CRKs were co-injected in a ratio of 2:1. To ensure uniform injections of transcripts, mixtures were made up to a standard volume as necessary with RNase-free water. Stage V and VI oocytes were isolated from mature Xenopus, and the follicular cell layer was digested with 2 mg ml$^{-1}$ collagenase (type 1A; Sigma-Aldrich) for 1 hr. Oocytes were incubated in 96-well plates with ND96 (in mM: 96 NaCl, 2 KCl, 1 $MgCl_2$, 1 $CaCl_2$, 5 HEPES-NaOH, pH 7.4) for overnight before injection. Around 25 ng of SLAC1 cRNA (with OST1) were injected into each oocyte. Injected oocytes were incubated in ND96 supplemented with gentamycin (5 mg ml$^{-1}$) at 18°C for 3 d before electrophysiological recordings. Whole-cell currents were recorded under voltage clamp using an Axoclamp 2B two-electrode clamp circuit (Axon Instruments). Measurements were performed in $I_{anion}$ measuring solution (in mM: 48 CsCl, 48 NaCl, 1 $MgCl_2$, 1 $CaCl_2$, 10 MES/TRIS, pH 5.6) for SLAC1 expressing oocytes. The voltage clamp protocols were set at: holding at 0 mV for 1 s, activation at +50 mV for 10 s, testing from +50 to −130 mV for 11 8 s cycles, and holding at 0 mV for 1 s (20 s cycles for SLAC1). All Xenopus experiments received ethical approval (Animal Ethics Application # S-2014–192, University of Adelaide).

## Chemilumiscence detection of ROS burst in plant leaf pieces

Leaves from healthy three-week old Arabidopsis plants were cut into small pieces using a leaf punch and floated on deionized water in individual wells of a 96-well plate overnight at room temperature. The next day, the deionized water was removed and replaced with 90 µl water and 10 µl of luminol master mix [200 µM luminol (Wako Chemicals, USA), 10 µg / ml horseradish peroxidase]. Flg22 elicitor, ABA, PAP or 0.1% ethanol (solvent control) was then gently added into the individual wells without disturbing the leaf pieces, with eight biological replicates per treatment. The plate was gently shaken and luminescence continuously measured in a plate reader (Tecan, USA) for 60 min.

## Ratiometric $[Ca^{2+}]_{cyt}$ imaging analysis

Three- to four-week-old plants transformed with yellow cameleon 3.6 (YC3.6) (*Mori et al., 2006*) were used to prepare epidermal peels. Leaves were attached with the abaxial side facing the glass bottom of the dish (Greiner, Germany) using medical adhesive (Hollister, USA) and upper cell layers were removed with a razor blade. Epidermal peels were incubated in the assay buffer (5 mM KCl, 10 mM MES-Tris (pH 5.6), 50 µM $CaCl_2$) for at least 1 hr before imaging. Initial $[Ca^{2+}]_{cyt}$ levels in guard cells were monitored for ≈20 min while the assay buffer was perfused with a peristaltic pump. PAP-mediated $[Ca^{2+}]_{cyt}$ level changes were monitored after the assay buffer containing 100 µM PAP was perfused. At the end of each experiment, 10 mM $CaCl_2$ was perfused into the bath medium to ensure the cells are capable of sensing $[Ca^{2+}]_{cyt}$ changes. Ratiometric imaging was conducted according to the method described in *Waadt et al. (2014)* using an Eclipse TE300 microscope equipped with a Cool SNAP HQ camera (Photometrics, USA), a Mac 2002 System automatic controller, and Cameleon filter set 71007A (D440/20, D485/40, D535/30; Chroma, USA). Images were acquired in intervals of 6 s.

## Imaging

Plant photographs were taken using the Lumix DMC-FZ5 camera (Panasonic, Japan). Examination of Arabidopsis leaves was performed using Cambridge S360 (SEM; 1987; Leica/Cambridge, Wetzlar, Germany). All images were saved in TIFF format and analyzed using ImageJ software. The stomatal index (SI) was calculated as SI = 100 x [S/(E+S)], where S is the number of stomata and E is the number of epidermal cells. Leica microscope (Leica Microsystems GMBH, Wetzlar, Germany) was used to observe changes in stomatal aperture after various chemical treatments. All images were taken using SPOT Advanced software (SPOTTM Imaging Solutions, MI, USA) for Windows version 4.0.9.

For GFP imaging the full length cDNA of SAL1 or of the small subunit of 1,5 Ribulose bisphosphate carboylase/oxygenase (SSU Rubisco) (the plastidic localization control) were cloned as a C-terminal GFP or RFP fusion, respectively, by gateway cloning under the control of the native SAL1 promoter or a 35S promoter respectively (*Murcha et al., 2007*). Both constructs were co-transformed into Arabidopsis cell suspension. Localization of GFP and RFP expression was conducted using an Olympus BX61 fluorescence microscope and imaged using the CellR imaging software as previously described (*Murcha et al., 2007*).

## Global transcript analyses

Analysis of the changes in transcript abundance between Col-0L*er*, *sal1*-8, *ost1*-2 and *ost1*-2 *sal1*-8 in the presence or absence of ABA was performed using Affymetrix Arabidopsis gene 1.0 ST arrays. Four week-old plants were sprayed with 20 µM ABA, the expected presence/absence of stomatal closure verified in all genotypes using infrared thermography, and the tissue samples were collected after 45 min of treatment for RNA analyses in biological triplicates using one leaf per sample per genotype per treatment. Total RNA was extracted using the guanidinium thiocyanate-phenol-chloroform extraction method (*Chomczynski and Sacchi, 1987*) using Trizol (Ambion), DNA removed using TURBO DNase, RNA quality verified using a Bioanalyzer (Agilent Technologies), and spectrophotometric analysis was performed to determine the A260:A280 and A260:A230 ratios. Labeling and hybridization to Affymetrix Arabidopsis Gene 1.0 Arrays was performed by the Ramaciotti Centre for Genomics (University of New South Wales, Sydney, Australia).

Pre-processing and statistical analysis of the Affymetrix Arabidopsis Gene 1.0 ST Arrays was performed in R (v3.1.0) using the Bioconductor packages oligo (v1.28.0) (*Carvalho and Irizarry, 2010*) and Limma (v3.20.1) (*Smyth, 2004*, *2005*) respectively; and also using Affymetrix Power Tools suite

(APT). Data quality checks and analysis were also performed using Partek Genomics Solution (PGS) software version 6.6 (default parameters; Partek, St. Louis, www.partek.com) and near identical results were obtained compared to the R workflow. For R analyses (shown in *Figure 3*, *Figure 3— figure supplement 2*, *Supplementary file 1*) raw cel files were read into R and the exon probe sets were background corrected, quantile normalized and expression values were summarized (using the median-polish) to the transcript level by calling the rma() function (*Irizarry et al., 2003a*, *2003b*) with the target='core' option (to summarize to gene level features with the highest annotation confidence/evidence). Affymetrix Power Tools were employed to calculate the detection above background (DABG) probability for each probe set. A gene was considered present if at least one half of its constituent exons were assigned a detection p<0.05, only genes that were detected as present in greater than half the sample replicates in at least one sample group were retained for further analysis. Probe sets were further filtered to remove unannotated and control probe sets as well as any probe set reported to cross-hybridize (only probes annotated as crosshyb = 1 by Affymetrix were retained), giving a final set of 13,780 detected genes.

## Gene expression analysis by qPCR

RNA was extracted from leaf tissue as described above, and reverse-transcribed into cDNA using the Invitrogen Superscript III cDNA Synthesis Kit (LifeTechnologies, USA) according to manufacturer's instructions. Exactly 1 µg total RNA was incubated with 50 pmol oligoDT primer (65°C, 10 min). Complementary DNA was then synthesised in a 20 µL reaction containing 1 mM dNTPs, 1 × first strand reaction buffer [250 mM Tris-HCl pH 3.8, 375 mM KCl, 15 mM $MgCl_2$], RNase inhibitor, and 100 units of Superscript III Reverse Transcriptase. The reaction was incubated at 50°C for 60 min, then the reaction stopped by heating (70°C, 15 min) and placing on ice. cDNA samples were stored at −20°C.

Gene expression was analysed on Roche LightCycler480 (Roche Diagnostics, Germany) by qPCR in 384-well plates, using a relative quantification method (*Ruijter et al., 2009*). For all experiments, three biological replicates were used per sample group or treatment, and each sample was analysed in three technical replicates. SybrGreen fluorescent intercalating dye (Roche Diagnostics, Germany) was used for quantification of relative transcript abundance. Gene-specific primers as defined in *Supplementary file 1* were used.

## In situ gene expression analysis

An in situ RT-PCR was performed directly on epidermal peels as described in *Athman et al. (2014)*. Epidermal peels were performed as described above; however, they were not fixed to a glass bottom chamber with the silicon adhesive. Instead, they were incubated in a fixation solution (2% formaldehyde solution, 63% ethanol, 5% acetic acid) on a microscope slide (Sigma Aldrich, USA). After fixation, the formaldehyde was removed by rinsing the peels with two wash solutions (A: 63% ethanol, 5% acetic acid; B. 0.01M $Na_2HPO_4$, 0.13M NaCl). Subsequently, the epidermal peels were DNAse treated using Ambion TURBO DNAse (LifeTechnologies, USA) as per the manufacturer's instructions. For first strand synthesis, SuperScript III (ThermoFisher Scientific, USA) was used with polyT/oligodT primers, as per the manufacturer's instructions. On these products a PCR reaction, incorporating DIG-11-dUTP, was performed using gene-specific primers and cycling conditions as outlined in *Supplementary file 1*.

To detect PCR products, peels were incubated with an anti-DIG antibody with a conjugated alkaline phosphatase (Roche, Switzerland), which binds to the DIG-labelled PCR products. Staining was achieved through incubation with a substrate of AP, BM purple (Roche, Switzerland), for 1 hr. Epidermal peels were then washed and mounted in 40% glycerol and viewed under a Leica DM5500B Bright Field Microscope, with attached camera, at 40X.

## Statistical analyses

Statistical analysis of differential gene expression was performed in R (version 3.1.3), where a linear model was fitted for the 2-factor design and coefficients determined for each of the 8 genotype x treatment factors (i.e. sample groups analyzed; equivalent to applying a classical genotype*treatment interaction model) using the lmFit function followed by empirical Bayes smoothing of the standard errors using eBays. Contrasts of interest were extracted and decideTests and topTable were

applied to determine differentially expressed transcripts between the different conditions/genotypes using the adjust.method="BH" option for the *Benjamini and Hochberg (1995)* method to adjust P-values for multiple comparisons. Transcripts were considered differentially expressed where the adjusted P-values were <0.05 (FDR 5%) and absolute fold-change >1.5. Gene-set enrichment tests were conducted using roast (*Wu et al., 2010*). For the gene-set analyses only probes mapping to a single TAIR10 gene were considered.

For biochemical and physiological experiments analyses of variance (ANOVA) were used to test for significant (p<0.05) differences between three or more sample groups for a particular treatment or time point. In limited cases where just two sample groups of interest were compared, the two-sample student's t-test assuming equal variance was used. Statistical procedures were carried out using the GraphPadInStat software (version 3.06), unless otherwise stated.

For stomatal aperture measurements, R (version 3.1.3) was used for statistical analyses, performing an ANOVA to test for significant differences (p<0.05) in relative pore area between treatments at each time point using the aov function, as part of the R 'stats' package, and modelling stomatal closure (X) with a nested model (timepoint nested within treatment: X ~ Treatment/Timepoint). Additionally, to look at rates of closure, a mixed-effect model was produced taking into account random effects between stomata nested with peels: X ~ Timepoint * Treatment + Peel.ID/Stomate.ID. This model observed only timepoints 10–25 min as this was considered to be the predominant period of stomata closure. This was conducted using the lmer function from the lme4 package (used alongside the lmerTest package which influences the tests performed by the lmer function). Importantly, for these analyses, each stomate was considered to represent a single biological unit. This assumption was confirmed from the mixed effect model produced, which demonstrated that the variation that occurred between stomata from a single epidermal peel (0.1779) was comparable to the variation observed between epidermal peels (0.2490) for a particular treatment.

## Acknowledgements

We thank J Schroeder (UC San Diego), S Assmann (Penn State Univ), I Small (Univ. Western Australia), as well as S Tyerman (Univ. Adelaide) for advice and critical comments; S Cutler (UC Riverside) for the *snrk* triple mutant; M Knight (Durham) for the *cas* mutant; Y Todoroki (Shizuoka Univ) for providing AS2; T Neeman, E Williams and K Murray (ANU) for help with statistical analyses; M Pitt for initial work on the double mutants; B Zhang and MR Blatt (Glasgow Univ) for plasmid DNAs of ion channels; WSU Confocal Bio-Imaging Facility for usage of confocal microscope; A Athman (Univ. Adelaide) for cloning and in-situ PCR assistance; S Yee (Australian National University) for work on the Col-0 background double mutants, Z-Y Wang (Australian National University) for assistance with the *ost1-2 xrn2*-1 *xrn3*-3 triple mutants, J Jimenez-Berni, R White and P Chandler (CSIRO) for help with infra-red thermography, microscopy, and the methods for ABA quantification, respectively. We received financial support from the ARC Centre of Excellence in Plant Energy Biology (CE140100008) and scholarships to WP (Thai Government and ScAWAKE), KXC (ANU), PAC (Grains Research Development Council), PBW (Meat and Livestock Australia), SYP (ANU), DG (GRDC, ANU), EET (ANU) and JP (HFSP Long-term fellowship). ZHC is supported by an ARC DECRA (DE140101143) and a 1000-Plan Project, China. A National Institutes of Health grant (GM060396) awarded to Julian Schroeder also funded research in this manuscript. Array data is available at NCBI GEO Database (GSE84997).

## Additional information

### Funding

| Funder | Grant reference number | Author |
|---|---|---|
| Australian Research Council | CE140100008 | Wannarat Pornsiriwong<br>Gonzalo M Estavillo<br>Kai Xun Chan<br>Estee E Tee<br>Diep Ganguly<br>Peter A Crisp<br>Su Yin Phua |

|  |  | Jiaen Qiu<br>Nazia Nisar<br>Arun Kumar Yadav<br>Christopher I Cazzonelli<br>Philippa B Wilson<br>Matthew Gilliham<br>Barry J Pogson |
| --- | --- | --- |
| ScAwake |  | Wannarat Pornsiriwong |
| Thai Government |  | Wannarat Pornsiriwong |
| The Australian National University |  | Kai Xun Chan<br>Estee E Tee<br>Su Yin Phua |
| Grains Research and Development Corporation |  | Diep Ganguly<br>Peter A Crisp |
| National Institutes of Health | GM060396 | Jiyoung Park |
| Human Frontier Science Program |  | Jiyoung Park |
| Meat and Livestock Australia |  | Philippa B Wilson |
| Australian Research Council | DECRA - DE140101143 | Zhong-Hua Chen |

The funders had no role in study design, data collection and interpretation, or the decision to submit the work for publication.

## Author contributions

WP, Conceptualization, Data curation, Formal analysis, Investigation, Project administration, Writing—review and editing, Drought stress assays, generation of mutants and genotyping, leaf physiological measurements, stomatal bioassays, led study design, data analysis and manuscript preparation; GME, Conceptualization, Data curation, Formal analysis, Supervision, Investigation, Project administration, Writing—review and editing, Generation of mutants and genotyping, leaf physiological measurements, protein localization and expression analysis, transcriptomics and gene expression analysis, led study design, data analysis and manuscript preparation, project administration; KXC, Conceptualization, Data curation, Formal analysis, Supervision, Investigation, Writing—original draft, Project administration, Writing—review and editing, Metabolite analysis, seed dormancy and germination analysis, stomatal bioassays and data analyses, transcriptomics and gene expression analysis, led study design, data analysis and manuscript preparation, project administration; EET, Data curation, Formal analysis, Investigation, Writing—review and editing, Electrophysiology, generation of mutants and genotyping, ROS assays, stomatal bioassays and data analyses, manuscript editing; DG, Data curation, Formal analysis, Investigation, Writing—review and editing, Stomatal bioassays and data analyses, gene expression analysis, manuscript editing; PAC, Data curation, Formal analysis, Writing—review and editing, Generation of mutants and genotyping, transcriptomics and gene expression analysis, manuscript editing; SYP, Data curation, Formal analysis, Investigation, Writing—review and editing, Metabolite analysis, seed dormancy and germination analysis, stomatal bioassay analysis, transcriptomics and gene expression analysis, manuscript editing; CZ, Formal analysis, Investigation, Electrophysiology, stomatal bioassays and data analyses; JQ, Formal analysis, Investigation, Electrophysiology and data analyses; JP, Formal analysis, Investigation, Writing—review and editing, Ca2+ imaging analysis, stomatal bioassays and data analyses, manuscript editing; MTY, Formal analysis, Investigation, Stomatal bioassays and data analyses; NN, Investigation, Drought stress assays, generation of mutants and genotyping, and data analyses; AKY, Investigation, Leaf physiological measurements, and data analyses; BS, Investigation, ROS assays, contributed to study design and interpretation of results; JR, Conceptualization, Contributed to study design and interpretation of results; CIC, Data curation, Formal analysis, Transcriptomics and gene expression analysis; PBW, Investigation, Drought stress assays and data analyses; MG, Formal analysis, Supervision, Contributed to study design and interpretation of results; Z-HC, Conceptualization, Data curation, Formal analysis, Supervision, Investigation, Writing—review and editing, Electrophysiology; ROS assays, stomatal bioassays and data analyses, led study design, data analysis and manuscript preparation; BJP, Conceptualization, Data curation, Supervision, Writing—original draft,

Project administration, Writing—review and editing, Led study design, data analysis and manuscript preparation, project administration

### Author ORCIDs
Kai Xun Chan, http://orcid.org/0000-0003-3554-7228
Peter A Crisp, http://orcid.org/0000-0002-3655-0130
Matthew Gilliham, http://orcid.org/0000-0003-0666-3078
Barry J Pogson, http://orcid.org/0000-0003-1869-2423

### Ethics

Animal experimentation: All experimentation involving Xenopus oocytes were performed in strict accordance to the University of Adelaide ethics committee guidelines. All Xenopus experiments received ethical approval (Animal Ethics Application # S-2014-192, University of Adelaide).

## Additional files

### Supplementary files

• Supplementary file 1. Global gene expression changes in response to ABA.

• Supplementary file 2: List of ABA signaling genes transcriptionally altered in *ost1-2 sal1*-8 +/- ABA.

### Major datasets

The following dataset was generated:

| Author(s) | Year | Dataset title | Dataset URL | Database, license, and accessibility information |
|---|---|---|---|---|
| Crisp PA, Estavillo GM, Pogson BJ | 2017 | Gene expression profiling of retrograde PAP-signaling and ABA-signaling mutants in response to ABA treatment | https://www.ncbi.nlm.nih.gov/geo/query/acc.cgi?acc=GSE84997 | Publicly available at the NCBI Gene Expression Omnibus (accession no: GSE84997) |

The following previously published dataset was used:

| Author(s) | Year | Dataset title | Dataset URL | Database, license, and accessibility information |
|---|---|---|---|---|
| Assmann SM | 2010 | Transcriptome analysis of Arabidopsis thaliana G protein subunit mutants in response to abscisic acid (ABA) | https://www.ncbi.nlm.nih.gov/geo/query/acc.cgi?acc=GSE19520 | Publicly available at the NCBI Gene Expression Omnibus (accession no: GSE19520) |

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
