## [Decision Letter]

Thank you for submitting your article "A chloroplast retrograde signal, PAP, acts as a secondary messenger in ABA signaling in stomatal closure and germination" for consideration by *eLife*. Your article has been reviewed by three peer reviewers, and the evaluation has been overseen by a Reviewing Editor and Detlef Weigel as the Senior Editor. The following individuals involved in review of your submission have agreed to reveal their identity: Robert Larkin (Reviewer #1); Markus Teige (Reviewer #2).

The reviewers have discussed the reviews with one another and the Reviewing Editor has drafted this decision to help you prepare a revised submission.

Summary:

In this manuscript Pornsiriwong et al. communicate a major conceptual advance to the plastid-to-nucleus signaling and drought tolerance demonstrating that PAP signaling promotes drought tolerance by restoring the ability of ABA to promote the closure of the stomata by activating a noncanonical ABA signaling mechanism. The reviewers thought that the genetics were done with a fair amount of rigor, suggesting that indeed the SAL1-PAP pathway is responsible for this effect and further studies show clearly that PAP itself acts as signal for stomatal closure. The array of phenotypic assays: measurements of ion fluxes, ROS production and large-scale transcriptional analysis was considered impressive. The reviewers concur with the conclusion that the chloroplastic PAP pathway can act in guard cells to prime or enhance ABA responses and also acts in seeds, which lack mature plastids, to regulate germination. The data may present clues as to how Ca signals are primed in guard cells. Overall, reviewers felt that the work offers some exciting new insights into how plastid derived signals may prime ABA and Ca signalling in guard cells

Essential revisions:

1) A question remains about if the signalling role of the PAP bypass is mediated by the CPKs 32 and 34 or CRKs2 and 8 – are those also upregulated in the *ost1-2 xrn2-1 xrn3-3* mutant? In their Discussion the authors seem also to favour this explanation of the transcriptional regulation of the "alternative kinases". Why was this not been tested directly?

2) The discussion about ABA needs to be more focused, less time spent on discussing elevated ABA and more of the mode of action of PAP in this bypass – does PAP directly activate downstream ABA signalling components, for example it might bind to some factors and regulate their activity – or does it mediate a transcriptional upregulation of other kinases, which are normally lower expressed and then take over this function?

3) Include the model in the main manuscript and to make the possibilities (above) a bit clearer.

4) The use of the dominant *abi1-1* in this work means that some results should be interpreted with caution. For example, previous work (Schroeder lab) gives precedent for PAP to prime ABA/Ca signalling, with ABA induced increases in Ca then bypassing the SNRKs. However, the same study also shows that PP2Cs (ABI1) can dephosphorylate SLAC1, specifically OST1 and CPK-mediated phosphorylation. This is problematic given that the allele used in this study is *abi1-1*, which is dominant and so might be expected to negate any PAP-CPK activation of SLAC1. However, this result (ABI1 dephosphorylation of SLAC1) is itself at odds with other literature and so is not conclusive.

5) The underlying reason for lowered stomatal conductance (Figure 1) needs to be investigated further. The conductance data in 1B might be due to significantly smaller stomata, which is supported by the higher cell densities and yet only a marginal per area increase in pore area per leaf area. Perhaps, stomatal bioassays (such as in 2A) using a range of ABA and Ca would help determine whether increase PAP in sal1-8 primes guard cells increases the sensitivity or dynamics to ABA and Ca. In this instance, the double mutants (*sal1-8* and *abi1-1* or *ost1-2*) are perhaps less informative as they are perturbed in a major branch of guard cell ABA signalling. However, here the data in Figure 7 is perhaps clearer and supportive of PAP increasing sensitivity to ABA (though ABA content of these seeds is not shown).

6) The potential for many of the observed responses to be due to PAPs interaction with the nuclear XRNs, rather than another role in Ca (dotted line in Figure 7—figure supplement 1) could be supported better by, for example, performing the same analyses with *ost1-2 xrn2 xrn3* as were done with the *ost1-2sal1-8* mutant to further support the discussion.

---

## [Author Response]

*Essential revisions:*

*1) A question remains about if the signalling role of the PAP bypass is mediated by the CPKs 32 and 34 or CRKs2 and 8 – are those also upregulated in the ost1-2 xrn2-1 xrn3-3 mutant? In their Discussion the authors seem also to favour this explanation of the transcriptional regulation of the "alternative kinases". Why was this not been tested directly?*

Thanks for this request.

Action: We have now tested the expression of the CDPKs in *ost1*-2 *xrn2*-1 *xrn3*-3. The new panel, Figure 5, shows that all four CDPKs are indeed significantly up-regulated in *ost1*-2 *xrn2*-1 *xrn3*-3, with the extent of up-regulation being strikingly similar to that in *ost1*-2 *sal1*-8. The up-regulation of these CDPKs is also accompanied by restoration of ABA-responsiveness at multiple levels in *ost1*-2 *xrn2*-1 *xrn3*-3 (see new Figure 4 and Essential revision #6 below), thus supporting our hypothesis that PAP is indeed mediating the bypass of OST1 through up-regulation of these CDPKs.

*2) The discussion about ABA needs to be more focused, less time spent on discussing elevated ABA and more of the mode of action of PAP in this bypass – does PAP directly activate downstream ABA signalling components, for example it might bind to some factors and regulate their activity – or does it mediate a transcriptional upregulation of other kinases, which are normally lower expressed and then take over this function?*

Done. We agree with the reviewer request and the additional requested data has aided in fulfilling the request.

Actions:

A) We have shortened the discussion on elevated ABA as suggested, while retaining the key points.

B) As stated in the response to Essential revision #1, the CDPKs capable of activating SLAC1 that are up-regulated in *ost1*-2 *sal1*-8 are also similarly up-regulated in *ost1*-2 *xrn2*-1 *xrn3*-3 (Figure 5). This up-regulation correlates with restoration of ABA responses at the level of ion fluxes, ROS burst, stomatal closure and increased leaf temperature in *ost1*-2 *xrn2*-1 *xrn3*-3 (Figure 4 and Essential revision #6).

B) We now clarify in the Discussion that based on our data, PAP acts in ABA signaling largely *via* XRNs to mediate a transcriptional up-regulation of kinases that are normally lowly-expressed. The possibility that PAP may target other proteins viadirect binding is also addressed in the Discussion and the model figure.

*3) Include the model in the main manuscript and to make the possibilities (above) a bit clearer.*

Done.

Action: The model is now included in the main manuscript as Figure 9, and the different possible modes of PAP action have been clarified both in the Discussion text and in the model.

*4) The use of the dominant abi1-1 in this work means that some results should be interpreted with caution. For example, previous work (Schroeder lab) gives precedent for PAP to prime ABA/Ca signalling, with ABA induced increases in Ca then bypassing the SNRKs. However, the same study also shows that PP2Cs (ABI1) can dephosphorylate SLAC1, specifically OST1 and CPK-mediated phosphorylation. This is problematic given that the allele used in this study is abi1-1, which is dominant and so might be expected to negate any PAP-CPK activation of SLAC1. However, this result (ABI1 dephosphorylation of SLAC1) is itself at odds with other literature and so is not conclusive.*

Done. This is a good point. It is possible that the complementation in *abi1*-1 does not proceed through CDPKs alone but additionally require other Ca^2+^ signaling proteins and/or transcription factors that are also up-regulated by PAP.

Action: We have now addressed the conflicting literature in the Discussion and explicitly consider whether up-regulation of CDPKs can complement *abi1*-1 through phosphorylation of SLAC1. In our view, the dominant effect of the ABI1-1 phosphatase is evident in the partial complementation of *abi1*-1 *sal1*-8 compared to the relatively complete complementation observed in *ost1*-2 *sal1*-8.

*5) The underlying reason for lowered stomatal conductance (Figure 1) needs to be investigated further. The conductance data in 1B might be due to significantly smaller stomata, which is supported by the higher cell densities and yet only a marginal per area increase in pore area per leaf area. Perhaps, stomatal bioassays (such as in 2A) using a of ABA and Ca would help determine whether increase PAP in range sal1-8 primes guard cells increases the sensitivity or dynamics to ABA and Ca. In this instance, the double mutants (sal1-8 and abi1-1 or ost1-2) are perhaps less informative as they are perturbed in a major branch of guard cell ABA signalling. However, here the data in Figure 7 is perhaps clearer and supportive of PAP increasing sensitivity to ABA (though ABA content of these seeds is not shown).*

Done. Our new results on both leaf peels and intact plants indicate that PAP accumulation in *sal1*-8 does lead to hypersensitivity to endogenous ABA; and this is more likely to be the cause of the observed lowered *g*_s_ in *sal1*-8 under control conditions rather than smaller stomata.

Action: We treated epidermal leaf peels and intact whole plants with a range of ABA concentrations. In both systems we observed that *sal1*-8 plants are indeed more sensitive to ABA compared to WT, with the stomata of the mutant closing more than WT in response to ABA in a dose-responsive manner (Figure 1—figure supplement 2).

*6) The potential for many of the observed responses to be due to PAPs interaction with the nuclear XRNs, rather than another role in Ca (dotted line in Figure 7—figure supplement 1) could be supported better by, for example, performing the same analyses with ost1-2 xrn2 xrn3 as were done with the ost1-2sal1-8 mutant to further support the discussion.*

Done. Our new results indicate that PAP can regulate stomatal closure through its interaction with nuclear XRNs.

Actions:

A) We now show that *ost1*-2 *xrn2*-1 *xrn3*-3 also has up-regulation of CDPK expression to similar extent to that observed in *ost1*-2 *sal1*-8 (Figure 5).

B) We now show that *ost1*-2 *xrn2*-1 *xrn3*-3 has restored ABA responsiveness at multiple levels (ROS burst, ion fluxes, stomatal closure and increased leaf temperature).

C) We have removed the dotted line to Ca^2+^ in the model. The possibility that PAP may target unknown proteins by direct binding (Essential revision #2) is also addressed in the model and in the Discussion.